# M3COTBENCH: BENCHMARK CHAIN-OF-THOUGHT OF MLLMS IN MEDICAL IMAGE UNDERSTANDING

**Juntao Jiang**[1*], **Jiangning Zhang**[1*], **Yali Bi**[2], **Jinsheng Bai**[1], **Weixuan Liu**[3], **Weiwei Jin**[4], **Zhucun Xue**[1], **Yong Liu**[1†], **Xiaobin Hu**[5], **Shuicheng Yan**[5]

[1] Zhejiang University, [2] University of Science and Technology of China, [3] East China Normal University, [4] Zhejiang Provincial People's Hospital, [5] National University of Singapore

## ABSTRACT

Chain-of-Thought (CoT) reasoning has proven effective in enhancing large language models by encouraging step-by-step intermediate reasoning, and recent advances have extended this paradigm to Multimodal Large Language Models (MLLMs). In the medical domain, where diagnostic decisions depend on nuanced visual cues and sequential reasoning, CoT aligns naturally with clinical thinking processes. However, current benchmarks for medical image understanding generally focus on the final answer while ignoring the reasoning path. Such opaque reasoning processes lack reliable bases for judgment, making it difficult to assist doctors in diagnosis. To address this gap, we introduce a new M3CoTBench benchmark specifically designed to evaluate the correctness, efficiency, impact, and consistency of CoT reasoning in medical image understanding. M3CoTBench features *1)* a diverse, multi-level difficulty dataset covering **24** examination types, *2)* **13** varying-difficulty tasks, *3)* a suite of CoT-specific evaluation metrics (correctness, efficiency, impact, and consistency) tailored to clinical reasoning, and *4)* a performance analysis of multiple MLLMs. M3CoTBench systematically evaluates CoT reasoning across diverse medical imaging tasks, revealing current limitations of MLLMs in generating reliable and clinically interpretable reasoning, and aims to foster the development of transparent, trustworthy, and diagnostically accurate AI systems for healthcare. Project page at https://juntaojianggavin.github.io/projects/M3CoTBench/.

## 1 INTRODUCTION

In recent years, Chain-of-Thought (CoT) reasoning has proven to be a transformative mechanism in enhancing the problem-solving capabilities of Large Language Models (LLMs) (Chu et al., 2024). By generating intermediate reasoning steps before arriving at a final answer, CoT improves transparency and structured decision-making in LLMs. Notable advancements include models like OpenAI's o1 (OpenAI, 2024b) and o3-mini (OpenAI, 2025c), which exhibit consistent, step-by-step logical reasoning across multi-turn interactions, and DeepSeek-R1 (DeepSeek-AI et al., 2025) that excels at decomposing complex tasks into fine-grained subtasks. Building on these successes, researchers have extended CoT to Multimodal Large Language Models (MLLMs) (Wang et al., 2025b), enabling joint processing of multiple modalities. Multimodal CoT frameworks now integrate visual and textual evidence into coherent multi-step explanations, with methods like Chain-of-Spot (Liu et al., 2024b), TextCoT (Luan et al., 2024), and DCoT (Jia et al., 2024) emphasizing region-of-interest analysis. Recent breakthroughs, such as OpenAI's o3 (OpenAI, 2024c) model, further demonstrate CoT's potential for image-based reasoning, while applications in healthcare, robotics, and autonomous driving highlight its versatility across domains.

In medical MLLMs, CoT reasoning is uniquely critical due to the complexity of medical image interpretation (Liu et al., 2024a). Clinicians rely on systematic diagnostic processes that involve iterative observation, verification against key features, and knowledge-based refinement. Explicit reasoning chains are essential to ensure safety, trustworthiness, and alignment with clinical guidelines. However, current benchmarks for medical image understanding focus solely on final-answer accuracy,

---

*Equal Contribution
†Corresponding Author

neglecting the quality of intermediate reasoning steps (Wu et al., 2024; Ye et al., 2024; Hu et al., 2024). For instance, state-of-the-art medical MLLM benchmarks evaluate visual question answering (VQA) performance without assessing *how* or *why* a model arrives at an answer. This gap limits the development of clinically reliable AI systems, as two models could produce identical answers through fundamentally flawed or incomparable reasoning paths. Such a lack of scrutiny over intermediate reasoning increases the risk of unnoticed errors, misdiagnoses, and overconfidence in models that appear accurate on surface metrics.

To address these challenges, we introduce a novel M3CoTBench benchmark that is designed to evaluate and standardize CoT reasoning in medical image interpretation. Specifically, we propose a novel curation pipeline, which includes *1)* the collection of diverse and high-quality medical images, *2)* automated data annotation, and *3)* manual review and calibration. By bridging the gap between medical diagnostic workflows and AI-driven reasoning, M3CoTBench not only facilitates transparent evaluation but also paves the way for developing clinically trustworthy MLLMs. Our contributions redefine evaluation standards in medical imaging, emphasizing the need for interpretable, step-by-step reasoning in high-stakes applications. Our work is guided by three core principles:

- **Diverse Medical VQA Dataset.** We curate a 1,079-image QA dataset spanning 24 modalities, stratified by difficulty and annotated with step-by-step reasoning aligned to clinical workflows.
- **Multidimensional CoT-Centric Metrics.** Evaluation criteria for reasoning correctness, efficiency, impact, and consistency, enabling granular performance analysis for various MLLMs.
- **Comprehensive Model Analysis.** We evaluate general-purpose and medical MLLMs by quantitative metrics and case studies, highlighting strengths and failure modes in clinical reasoning to guide future improvements.

## 2 RELATED WORK

### 2.1 MULTIMODAL LARGE LANGUAGE MODELS

Inspired by recent advances in LLMs like LLaMA (Touvron et al., 2023) and GPT (Ouyang et al., 2022), MLLMs extend text-centric architectures by embedding visual features into the latent language space, enabling diverse image-grounded text generation. The LLaVA-OneVision (Li et al., 2024) family combines large-scale image/video corpora with instruction finetuning to excel across single-image, multi-image, and video tasks. LLaVA-CoT (Xu et al., 2024) introduces a multistage prompting strategy incorporating summarization, visual analysis, reasoning, and conclusion. Qwen-3-VL (Bai et al., 2025) employs enhanced interleaved-MRoPE for improved spatial-temporal modeling, DeepStack integration to leverage multi-level ViT features for tighter vision-language alignment, and text-based time alignment for precise video temporal grounding, enabling stronger multimodal understanding and long-context reasoning. InternVL 3.5 (Wang et al., 2025a) substantially improves reasoning capability and inference efficiency through cascade reinforcement learning and efficient vision-language deployment. Closed-source GPT-4o (OpenAI, 2024a) exemplifies integration of real-time vision, audio, and text reasoning. GPT-5 (OpenAI, 2025b) intelligently routes queries to either fast responses or deeper reasoning, delivering smarter, more accurate, and more useful performance across domains. Claude-Sonnet-4.5 (Anthropic, 2025) and Gemini 3 (Google DeepMind, 2024) also show remarkable performances. In medicine, specialized MLLMs adapt these techniques to clinical data: Med-Flamingo (Moor et al., 2023) augments Flamingo (Alayrac et al., 2022) with medical image-text pretraining for few-shot VQA; LLaVA-Med (Li et al., 2023) aligns visual content with biomedical concepts using PubMed captions and GPT-4 instructions; Lingshu (Team et al., 2025) is a generalist multimodal medical foundation model that unifies medical image-text understanding and reasoning across diverse clinical tasks. MedGemma (Sellergren et al., 2025) is a long-context medical vision-language model built on Gemma, optimized for clinical image analysis and medical QA. Recent advances in MLLMs also catalyze the development of medical agents capable of assisting in diagnosis, decision support, and workflow integration (Hu et al., 2025).

### 2.2 MEDICAL MULTIMODAL BENCHMARKS

Medical multimodal benchmarks evaluate how well MLLMs interpret and reason over clinical imaging data. VQA-RAD (Lau et al., 2018) is an early radiology VQA dataset with clinician-annotated QA

pairs. PathVQA (He et al., 2020) extends VQA to pathology by pairing textbook and digital pathology images with expert-reviewed questions. SLAKE (Liu et al., 2021) offers English-Chinese radiology QA enriched with semantic labels linked to a structured medical knowledge base. FMBench (Wu et al., 2024) is the first to systematically assess fairness in MLLMs, incorporating clinical tasks, demographic-aware evaluation, and a novel disparity metric. Quilt-VQA (Seyfioglu et al., 2024) targets histopathology VQA using real-world images and curated questions. OmniMedVQA (Hu et al., 2024) aggregates diverse datasets spanning multiple modalities and anatomy, requiring models to integrate heterogeneous inputs and justify their answers. GMAI-MMBench (Ye et al., 2024) unifies 284 global datasets into a large-scale multimodal QA benchmark covering a broad range of clinical scenarios. Med-CMR (Gong et al., 2025) is a fine-grained evaluation benchmark specifically designed to measure how models integrate medical images with clinical reasoning to perform complex multimodal reasoning. However, they often lack annotations for intermediate reasoning steps, limiting their effectiveness in assessing CoT-style clinical inference in complex medical scenarios.

## 2.3 CoT-Related MLLM Benchmarks

Research on reasoning in multimodal models has advanced through several dedicated benchmarks. Visual-CoT (Shao et al., 2024) introduces a large-scale dataset of image-QA pairs, augmented with region annotations and step-by-step rationales, along with a multi-turn reasoning pipeline for interpretable, region-focused CoT tasks. M$^3$CoT (Chen et al., 2024b) provides a comprehensive benchmark spanning diverse domains and requiring complex multi-step visual-textual reasoning. MME-CoT (Jiang et al., 2025) extends this line of work by contributing high-quality data across six domains and proposing three novel metrics to assess CoT quality, robustness, and efficiency. CoMT (Cheng et al., 2025) introduces a benchmark that requires both multimodal inputs and outputs to evaluate the visual reasoning abilities of MLLMs, addressing the limitations of traditional text-only outputs in multimodal CoT tasks. While these benchmarks have advanced CoT reasoning in natural image domains, resources remain scarce in the medical field, where rigorous diagnostic reasoning, interpretability, and domain expertise are essential. This gap underscores the need for medically grounded benchmarks that can assess step-by-step clinical inference in multimodal settings.

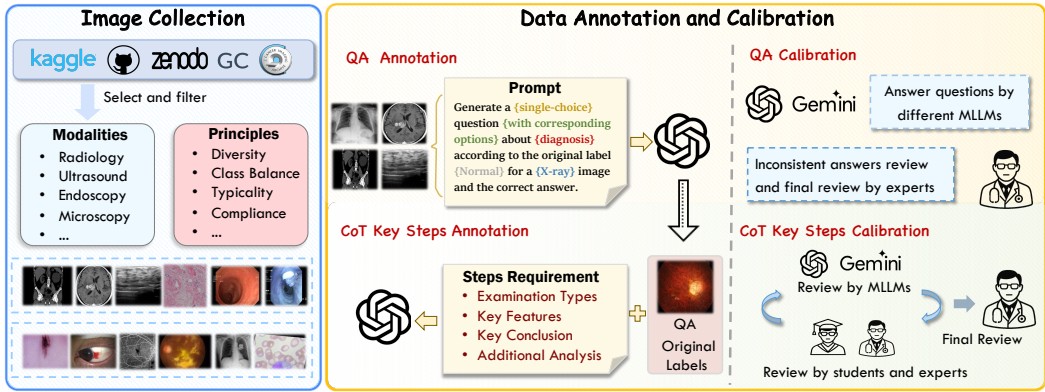

Figure 1: **Curation of M3CoTBench benchmark** that encompasses three sections: *1)* carefully curated medical images from various public sources, *2)* multi-type and multi-difficulty QA generation via LLMs and expert calibration, and *3)* structured annotation of key reasoning steps aligned with clinical diagnostic workflows.

## 3 Curation of M3CoTBench

The collection of images, construction of QA pairs, the annotation of key CoT steps, and manual review/calibration are carefully designed, as shown in Figure 1.

### 3.1 Data Collection

All images in M3CoTBench are sourced from public datasets, with selection guided by principles of diversity, typicality, class balance, and compliance.

- **Diversity.** Images are collected from 55 public medical datasets, encompassing diverse imaging modalities, examination types, and anatomical regions (Table A1), with broad geographical coverage (Figure A3) and diverse temporal ranges of publishing.

- **Typicality.** To ensure large intra-dataset variance, image features are extracted by Biomed-CLIP (Zhang et al., 2023), and a semantically distinct subset is selected by maximizing the minimum pairwise feature distance.

- **Class Balance.** Each public dataset comprises multiple subcategories, and we actively maintain a balanced representation of these subcategories during collection through careful inspection of the original labels.

- **Compliance.** Datasets with usage restrictions or labeled as "no derivatives" are excluded, addressing compliance issues often neglected in prior benchmarks.

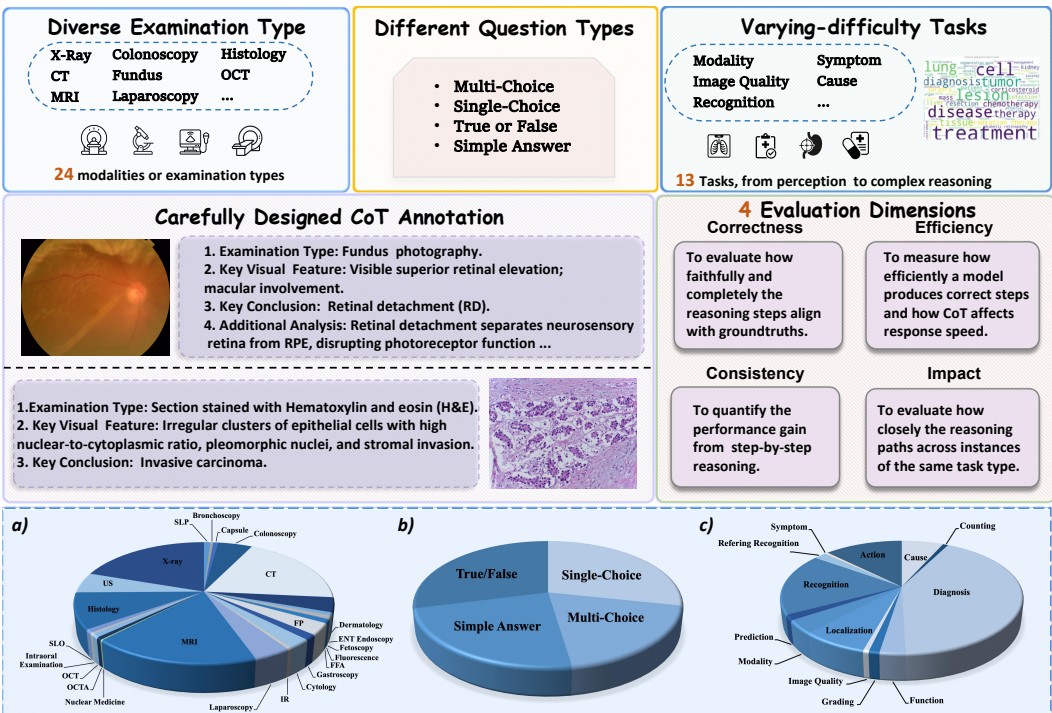

Figure 2: **Overview of M3CoTBench benchmark. Top:** The benchmark covers 24 imaging modalities/examination types, 4 question types, and 13 clinical reasoning tasks. **Middle:** CoT annotation examples and 4 evaluation dimensions. **Bottom:** The distribution of image-QA pairs across *a)* modalities, *b)* question types, and *c)* tasks.

## 3.2 DATA ANNOTATION AND CALIBRATION

**Question-Answer Pairs Generation.** We employ a unified pipeline for generating QA pairs, with all questions and candidate answers initially fully generated by GPT-4o, and subsequently calibrated by three different MLLMs and human experts to ensure the validity of the questions and the correctness of the answers.

- **Conversion of Existing Datasets.** We apply different strategies to different public datasets according to their diverse original purposes. Starting with existing QA pairs from public VQA and image classification datasets, we use GPT-4o to rewrite them into more diverse formats, such as single-choice, multiple-choice, true/false, and short-answer questions. For segmentation datasets, we concatenate the raw image with its corresponding mask and ask targeted questions about the masked region; for object detection datasets, we generate spatial questions, such as requesting a rough anatomical location or estimating bounding box coordinates; and for some image quality assessment and disease grading tasks, we present paired images and formulate comparative questions.

- **Generation of Inference-driven Medical Questions.** To enrich the complexity of QA tasks and better support reasoning capabilities, we provide GPT-4o with the original label and prompt it to generate questions with corresponding answer options grounded in that information. For example, given a slit lamp image labeled "severe keratitis with corneal ulcer", GPT-4o is prompted to create a multi-choice question about causes, such as "What might be the cause of this condition? (Select all that apply)", with answer options including bacterial, viral, fungal infections, trauma, allergic reactions, etc. The correct answers align with clinically relevant causes associated with the diagnosis. This approach introduces hierarchical difficulty and inference-driven tasks that extend beyond surface-level recognition, fostering more in-depth medical reasoning.

- **AI and Human Expert Calibration Process.** For calibration, we leverage three different MLLMs to answer each image-question pair independently. If any MLLM's response differs from the initially generated answer, a human expert, an experienced doctor, intervenes to make the final judgment. Additionally, the expert reviews all images and QA pairs comprehensively to perform a final quality check and calibration. This combined AI-human validation ensures high accuracy and reliability of the dataset.

**Rationale for the Step Design.** Our CoT steps are derived from clinician interviews and established theories of medical reasoning. They consist of: (1) confirming the image's nature (modality/examination type), (2) identifying key visual features, (3) drawing diagnostic conclusions, and (4) providing further medically informed analysis.

- **Validation via Doctor Interviews.** Before designing the CoT steps, we interview clinicians, radiologists, and sonographers from five hospitals. Most clinicians describe their workflow as: identify the imaging modality, observe key features, draw core conclusions, and then perform additional analyses such as etiology or treatment planning. Some clinicians note that intuition may guide an initial hypothesis, which is then verified through feature inspection. These findings support our chosen steps as both sufficient and necessary for medical reasoning.

- **Theoretical Support from Medical Cognition.** Our CoT design draws on established cognitive models. (1) Hypothetico-deductive reasoning (Elstein et al., 1978): Clinicians generate and iteratively test hypotheses; our steps follow this natural cycle. (2) Pattern recognition (Norman et al., 2007): Experienced doctors rapidly spot salient imaging patterns; our early focus on key features reflects this process. (3) Dual-process theory (Arvai, 2013): Intuitive and analytical reasoning interact; our annotations capture this by allowing preliminary intuitive judgments followed by feature-based verification and further analysis.

**CoT Key Steps Generation.** To ensure effective CoT in medical VQA that mirrors clinicians' cognitive workflow from perception to judgment, we first leverage MLLMs to annotate CoT key steps, which then undergo repeated cycles of review, feedback, and revision by medical experts and students before senior experts confirm the final CoT annotations.

- **MLLM-Based Annotation.** For each image-QA instance, we provide GPT-4o and Gemini-2.5-Pro with the image, the question, the answer, and any relevant contextual information from the original annotations. For example, underlying labels used to construct the question itself, complex questions about treatment, causes, prediction, or function are often derived from simpler labels, such as disease type, which are also provided as input. Additionally, the model generates reasoning steps following an expert-designed four-step clinical structure: (1) confirming the nature of the image, such as the imaging modality and examination type; (2) identifying key visual features; (3) drawing diagnostic conclusions, including the relevant disease, organ, or tissue; and (4) providing additional analysis based on medical knowledge, such as treatment strategies or associated symptoms. It is worth noting that we condition the model by specifying the expected reasoning steps based on the task type. For instance, modality questions omit steps 3 and 4, while diagnostic questions skip step 4. GPT-4o and Gemini 2.5 Pro then generate the corresponding key reasoning steps accordingly. Finally, the final results are generated again by GPT-4o, which integrate initial annotation information from both GPT-4o and Gemini 2.5 Pro.

- **AI and Human Expert Calibration Process.** To ensure high-quality and medically reliable annotations, we adopt a multi-stage human-AI collaborative verification process: **(1) Initial Student Review:** A medically trained student manually reviews model- or human-generated annotations, correcting factual, spelling, and formatting errors, and filling in missing key information. Uncertain cases are discussed with experts. **(2) Automated Multi-model Checking:** The image,

question, and reasoning steps are validated using GPT-4o. **(3) Expert Review on Model Flags:** Any reasoning step flagged as "potentially incorrect" by any model is sent to an expert in the relevant imaging modality for manual review. **(4) Consensus Resolution:** When experts identify issues, the involved experts and student reviewers hold brief online meetings or asynchronous discussions to resolve disagreements. Three such meetings and multiple asynchronous discussions are held. Annotations are updated based on consensus. **(5) Final Expert Read-through:** Experts conduct a final pass on each sample to ensure that the image, question, reasoning steps, and answer are medically correct, consistent, and compliant with benchmark standards.

## 3.3 Data Composition and Categorization

As shown in Figure 2, M3CoTBench includes diverse image-QA pairs with multiple question formats and task types of varying difficulty. It covers a broad range of imaging modalities and examination types across several categories. Tasks span from basic perception to advanced medical reasoning, enabling comprehensive evaluation of MLLMs.

**QA Types.** We include four question formats: single-choice, multiple-choice, true/false (judgment), and short-answer, spanning 13 task types with varying difficulty levels.

**Examination Types.** The dataset encompasses 24 imaging modalities and examination types, which can be organized into six major categories: ophthalmic imaging, radiology, endoscopy, microscopy, ultrasound-based examinations, and surface-level inspections. Representative modalities within these categories include slit lamp photography (SLP), fundus photography (FP), optical coherence tomography (OCT), optical coherence tomography angiography (OCTA), scanning laser ophthalmoscopy (SLO), fundus fluorescein angiography (FFA), X-ray, computed tomography (CT), magnetic resonance imaging (MRI), ultrasound (US), infrared reflectance (IR), nuclear medicine, fetoscopy, laparoscopy, colonoscopy, gastroscopy, capsule endoscopy, bronchoscopy, ENT endoscopy, histology, cytology, fluorescence microscopy, dermoscopy, and intraoral examination.

**Task Types.** To thoroughly assess the reasoning ability of MLLMs, we design questions spanning a broad spectrum of clinical tasks, including: Examination Type, Image Quality, Recognition, Referring Recognition, Localization, Diagnosis, Grading, Prediction, Function, Symptom, Counting, Cause, and Action. These categories range from low-level perception tasks to high-level clinical reasoning. Such a taxonomy is constructed to test MLLMs' ability to bridge the gap between visual perception and domain knowledge reasoning, challenging both their vision-language alignment and medical understanding. Some example image-question pairs can be seen in Figure 3.

Table 1: **Criterion comparison for current benchmarks.** ✓: Satisfied. ✗: Unsatisfied.

| Dataset | #Img/#QA | Exam. Type | Task | Question Type | CoT Annotation | Eval. Dimension | | | | |
|---|---|---|---|---|---|---|---|---|---|---|
| | | | | | | Acc. | Corr. | Imp. | Eff. | Cons. |
| VQA-RAD (Lau et al., 2018) | 315 / 3515‡ | 3 | 8 | 2 | ✗ | ✓ | ✗ | ✗ | ✗ | ✗ |
| SLAKE (Liu et al., 2021) | 642 / 14028‡ | 3 | 10 | 2 | ✗ | ✓ | ✗ | ✗ | ✗ | ✗ |
| Quilt-VQA (Seyfioglu et al., 2024) | 985 / 1283 | 1 | 8 | 2 | ✗ | ✓ | ✗ | ✗ | ✗ | ✗ |
| OmniMedVQA (Hu et al., 2024) | 118010 / 127995 | 12† | 5 | 3 | ✗ | ✓ | ✗ | ✗ | ✗ | ✗ |
| GMAI-MMBench (Ye et al., 2024) | - / 25831 | **38**† | 6 | 2 | ✗ | ✓ | ✗ | ✗ | ✗ | ✗ |
| M3CoTBench | 1079 / 1079 | 24 | **13** | **4** | ✓ | ✓ | ✓ | ✓ | ✓ | ✓ |

† The way of classifying modalities differs from this paper. ‡ The number of images and QA pairs also includes training sets.

## 4 Evaluation Suite of M3CoTBench

We evaluate CoT reasoning based on four aspects: correctness, efficiency, impact, and consistency. Here, correctness measures whether the generated reasoning steps are accurate; efficiency reflects the additional inference time introduced by reasoning; impact quantifies the overall effect of reasoning on answer accuracy compared to direct prediction without reasoning; and consistency assesses whether similar tasks tend to follow similar reasoning paths.

**Evaluation of Reasoning Correctness.** To comprehensively evaluate the accuracy of the model's reasoning steps, we quantify the alignment between the generated reasoning sequence and expert-annotated reasoning paths. Specifically, we compute the following metrics:

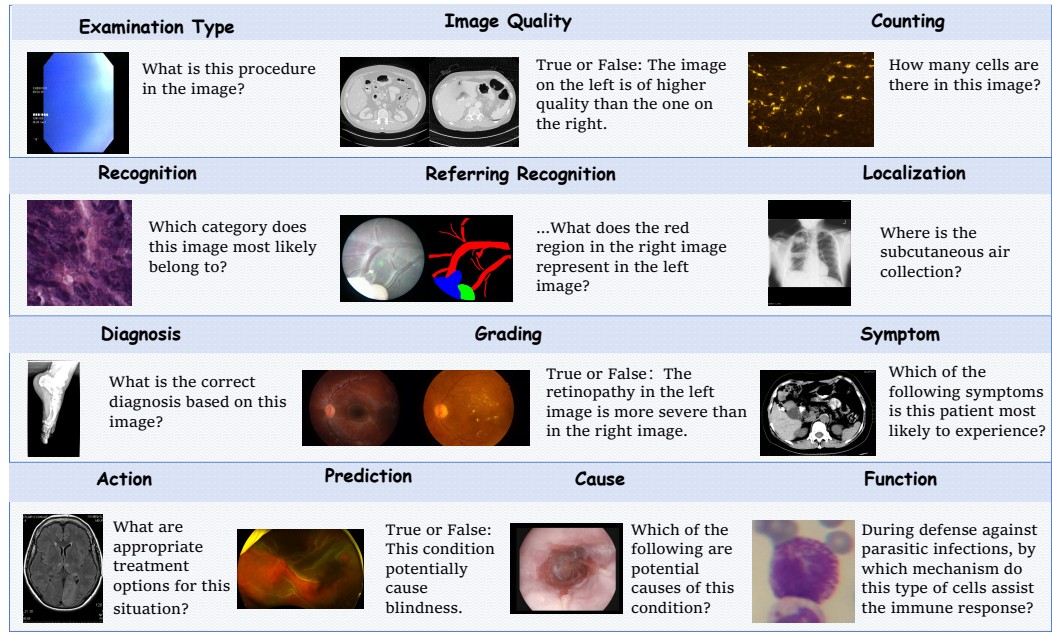

Figure 3: **Example image-question pairs for 13 tasks in M3CoTBench**, including identifying examination types, image quality assessment, recognition, referring recognition, counting, localization, diagnosis, grading, symptom identification, clinical action planning, prediction, functional understanding, and causal reasoning.

$$\text{Avg Precision} = \frac{1}{N} \sum_{i=1}^{N} |\mathcal{R}^{(i)} \cap \mathcal{A}_{k^*}^{(i)}| / |\mathcal{R}^{(i)}|, \quad \text{Avg Recall} = \frac{1}{N} \sum_{i=1}^{N} |\mathcal{R}^{(i)} \cap \mathcal{A}_{k^*}^{(i)}| / |\mathcal{A}_{k^*}^{(i)}|. \quad (1)$$

Here, $\mathcal{R}$ denotes the set of reasoning steps generated by the model, $\{\mathcal{A}_k\}$ represents all annotated gold reasoning paths for a given question, $N$ is the number of examples. Since multiple valid reference paths may exist, we choose the reference $\mathcal{A}_{k^*}$ with the highest overlap with $\mathcal{R}$. Precision measures the proportion of model-generated steps that are correct, while recall quantifies the coverage of reference reasoning steps. The F1 score is used to combine both aspects to provide a holistic evaluation of CoT correctness.

**Evaluation of Reasoning Efficiency.** CoT reasoning often introduces significant computational overhead due to longer generated sequences. Excessively verbose CoT outputs increase inference time and memory consumption, reducing practical usability in real-world applications. To evaluate reasoning efficiency, we compute the number of correct reasoning steps per unit time. Formally,

$$E = \sum_{i=1}^{N} \left| \mathcal{R}^{(i)} \cap \mathcal{A}_{k^*}^{(i)} \right| / T_{\text{CoT}}. \quad (2)$$

A higher $E$ indicates more accurate reasoning steps per unit time, reflecting more efficient reasoning. Then we define the average inference latency impact $L$ as the ratio between the total CoT inference time and total direct inference time, divided by the number of examples: $L = T_{\text{CoT}}/T_{\text{direct}}$, where $T_{\text{CoT}}, T_{\text{direct}}$ are the total inference times with and without CoT, respectively. A larger $L$ value indicates a greater average latency overhead. By jointly considering $E$ and $L$, we can better benchmark the trade-offs between interpretability and computational cost in CoT-enabled models.

**Evaluation of Reasoning Impact.** To quantify the benefit of generating step-by-step reasoning over directly producing the final answer, we define the reasoning impact metric as the difference in answer accuracy between the two approaches. Let $\text{Acc}_{\text{step}}$ denote the accuracy of the model when generating

answers with intermediate reasoning steps, and $\text{Acc}_{\text{direct}}$ denote the accuracy when generating answers directly without explicit reasoning. The reasoning impact $I$ is computed as: $I = \text{Acc}_{\text{step}} - \text{Acc}_{\text{direct}}$.

A positive value of $I$ indicates that step-by-step reasoning improves answer correctness, demonstrating the effectiveness of CoT generation in enhancing model performance. Conversely, a negative or zero value suggests that the reasoning steps do not provide additional benefit or may even degrade the correctness. This metric offers a straightforward way to assess whether incorporating explicit reasoning contributes meaningfully to the model's accuracy.

**Evaluation of Reasoning Consistency.** Structured and task-specific reasoning pathways are fundamental for interpretability and play a vital role in ensuring reproducibility, transparency, and trustworthiness in high-stakes medical decision-making. However, existing evaluation metrics often treat reasoning steps as unordered elements. To address this gap, we introduce a path consistency metric that explicitly evaluates how closely the reasoning paths for instances within the same task type align. We compute this score independently for each of the thirteen tasks and then average the results. For task $t$ with $N$ examples, represent each generated reasoning path $P_i^{(t)}$ as an ordered sequence of step categories (e.g., *modality*, *feature*, *diagnosis*, *additional analysis*). To evaluate path consistency, we first select the reference path by maximizing its average similarity with all generated paths:

$$P^{(t)} = \arg\max_P \sum_{i=1}^{N} \text{sim}\big(P, P_i^{(t)}\big), \quad \text{sim}\big(P, P_i^{(t)}\big) = |\text{LCS}(P, P_i^{(t)})|/\max(|P|, |P_i^{(t)}|). \quad (3)$$

The task-level consistency score is then defined as the average similarity between each path and the canonical reference:

$$C_{\text{path}}^{(t)} = 1/N \sum_{i=1}^{N} \text{sim}(P^{(t)}, P_i^{(t)}). \quad (4)$$

Average score over all tasks: $C_{\text{path}} = 1/M \sum_{t=1}^{M} C_{\text{path}}^{(t)}$, where $M = 13$. A higher $C_{\text{path}} \in [0, 1]$ indicates that the model has strong structural stability in its CoT.

## 5 EXPERIMENTS

### 5.1 EXPERIMENT SETUP

**Evaluation Models.** We select top-performing MLLMs for comprehensive CoT evaluation. We evaluate open-source models such as LLaVA-CoT(11B) (Xu et al., 2025), InternVL3.5(8B, 30B) (Wang et al., 2025a), Qwen3-VL-Instruct(8B, 30B) (Bai et al., 2025), Qwen3-VL-Thinking(8B, 30B) (Bai et al., 2025). We also include closed-source GPT-4.1 (OpenAI, 2025a), GPT-5 (OpenAI, 2025b), Gemini 2.5 Pro (Google DeepMind, 2024), and Claude-Sonnet-4.5 (Anthropic, 2025) as strong baseline models. Finally, we evaluate some models specifically designed for the medical domain, like LLaVA-Med (7B) (Li et al., 2023), HuatuoGPT-Vision-7B-Qwen2.5VL (Chen et al., 2024a), HealthGPT(3.8B) (Lin et al., 2025), Lingshu (7B, 32B) (Team et al., 2025) and MedGemma (4B, 27B) (Sellergren et al., 2025).

**Implementation Details.** We define the CoT prompt as: *Please generate a step-by-step answer, including all intermediate reasoning steps, and provide the final answer at the end.* The direct prompt is defined as: *Please directly provide the final answer without any additional output.* For all experiments, the batch size is set to 1 to ensure independent processing of each sample, and the temperature is uniformly set to 0.1. For evaluation, we use GPT-4o and LLaMA-3.3-70B-Instruct-Turbo (Grattafiori et al., 2024) and Gemini 2.5 Pro for assessment criteria. All local inference experiments are conducted on a server with AMD GPUs. APIs are used for closed-source MLLMs.

### 5.2 QUANTITATIVE RESULTS

The experimental results can be seen in Table 2, from which there are some interesting findings:

**Correctness. (1) Closed-source vs. Open-source Models:** Across models, closed-source systems do not exhibit a uniform advantage over open-source ones in terms of CoT-ground truth alignment. A

Table 2: **M3CoTBench results for MLLMs.** ↑(↓): the higher(lower) the better. $F1$, $P$, $R$: the average of F1 score(%), Precision(%), and Recall(%). $\text{Acc}_{\text{direct}}$ and $\text{Acc}_{\text{step}}$: accuracy(%) of generated answers by directly and CoT. $I$, $E$ and $L$, and $C_{\text{path}}$(%): Impact, Efficiency, Latency, and Consistency score, respectively. Optimal / sub-optimal results are highlighted in **bold** / underline.

| Model | Correctness | | | Impact | | | Efficiency | | Consistency |
|---|---|---|---|---|---|---|---|---|---|
| | $F1(\uparrow)$ | $P(\uparrow)$ | $R(\uparrow)$ | $\text{Acc}_{\text{direct}}$ | $\text{Acc}_{\text{step}}$ | $I(\uparrow)$ | $E(\uparrow)$ | $L(\downarrow)$ | $C_{\text{path}}(\uparrow)$ |
| *Open-source MLLMs* | | | | | | | | | |
| LLaVA-CoT (Xu et al., 2025) | 49.80 | 54.08 | 46.15 | 40.13 | 36.75 | -3.38 | 0.06 | 1.56 | 77.02 |
| InternVL3.5-8B (Wang et al., 2025a) | 56.48 | 60.61 | 52.88 | 56.81 | 53.61 | -3.20 | 0.10 | 18.27 | 71.65 |
| InternVL3.5-30B (Wang et al., 2025a) | 59.42 | 62.15 | 56.92 | **63.81** | 57.65 | -6.16 | 0.03 | 16.68 | 76.30 |
| Qwen3-VL-Instruct-8B (Bai et al., 2025) | 55.17 | 52.74 | 57.84 | 51.30 | 46.62 | -4.68 | 0.04 | 93.94 | 82.65 |
| Qwen3-VL-Instruct-30B (Bai et al., 2025) | 59.15 | 56.13 | 62.51 | 54.63 | 51.39 | -3.24 | 0.03 | 35.63 | 83.01 |
| Qwen3-VL-Thinking-8B (Bai et al., 2025) | 59.87 | 59.84 | 59.91 | 48.33 | 52.83 | **+4.50** | 0.02 | 2.79 | 76.91 |
| Qwen3-VL-Thinking-30B (Bai et al., 2025) | 62.15 | 63.34 | 61.01 | 51.95 | 55.47 | +3.52 | 0.02 | 1.15 | 76.02 |
| *Closed-source MLLMs* | | | | | | | | | |
| GPT-4.1 (OpenAI, 2025a) | 60.76 | 58.32 | 63.42 | 56.77 | 58.11 | +1.34 | 0.17 | 5.08 | 81.31 |
| GPT-5 (OpenAI, 2025b) | 55.13 | 64.15 | 48.34 | 58.76 | 58.29 | -0.47 | 0.06 | **1.10** | 65.39 |
| Gemini 2.5 Pro (Google DeepMind, 2024) | **66.07** | 62.48 | **70.10** | 60.24 | **60.10** | -0.14 | 0.10 | 1.52 | 82.00 |
| Claude-Sonnet-4.5 (Anthropic, 2025) | 56.50 | 53.62 | 59.71 | 51.34 | 51.07 | -0.27 | 0.15 | 2.69 | **85.22** |
| *Medical-Specific MLLMs* | | | | | | | | | |
| LLaVA-Med (7B) (Li et al., 2023) | 30.51 | 36.33 | 26.30 | 29.38 | 29.29 | -0.09 | **0.35** | 3.22 | 72.68 |
| HuatuoGPT-Vision (7B) (Chen et al., 2024a) | 49.45 | 51.17 | 47.85 | 41.89 | 34.94 | -6.95 | 0.21 | 5.92 | 73.19 |
| HealthGPT (3.8B) (Lin et al., 2025) | 32.56 | 47.27 | 24.83 | 44.11 | 42.03 | -2.08 | 0.06 | 15.36 | 67.72 |
| Lingshu-7B (Team et al., 2025) | 57.57 | 63.96 | 52.34 | 50.00 | 42.08 | -7.92 | 0.30 | 8.37 | 74.83 |
| Lingshu-32B (Team et al., 2025) | 59.16 | **65.68** | 53.82 | 51.81 | 44.95 | -6.86 | 0.21 | 10.87 | 71.47 |
| MedGemma-4B (Sellergren et al., 2025) | 48.13 | 50.29 | 46.14 | 43.33 | 41.29 | -2.04 | 0.05 | 20.61 | 74.03 |
| MedGemma-27B (Sellergren et al., 2025) | 50.98 | 48.33 | 53.81 | 46.06 | 45.46 | -0.60 | 0.03 | 23.71 | 82.55 |

notable example is GPT-5, which achieves relatively high Precision but substantially lower Recall. This pattern arises because GPT-5 frequently bypasses step-by-step reasoning and directly outputs final answers even under CoT instructions. In contrast, GPT-4.1 and Gemini 2.5 Pro show strong and balanced P/R/F1 scores, indicating that they reliably produce complete and structured reasoning chains. This comparison suggests that adherence to CoT instruction, rather than model openness, is the dominant factor in CoT quality. **(2) Thinking Models vs. Instruction Models:** Within the Qwen3-VL family, the Thinking variants consistently outperform the Instruct variants, demonstrating the benefit of explicitly modeling multi-step reasoning. The Thinking models are trained to externalize intermediate decisions, resulting in more complete coverage of annotated steps. **(3) Large vs. Small Models within the Same Series:** Comparisons within the same model family show that larger models generally achieve higher F1 scores than their smaller counterparts. Beyond raw capacity, larger models exhibit greater stability in multi-step reasoning, making them less prone to skipping steps, collapsing reasoning, or introducing spurious assumptions. **(4) Performance of Medical-Specific MLLMs:** Medical-specialized MLLMs do not consistently outperform general-purpose MLLMs in CoT-GT alignment. Many medical MLLMs show moderate Precision but lower Recall. This suggests that domain specialization alone does not guarantee high-quality CoT generation.

**Efficiency.** After introducing CoT prompts, the inference time for all models increases, highlighting that the computational overhead of step-by-step reasoning is a critical factor affecting overall system latency. Models exhibit markedly different latency behaviors in response to this slowdown. Models like LLaVA-CoT and the Qwen3-VL-Thinking series inherently generate step-by-step outputs; thus, adding a CoT prompt does not significantly alter the sequence length, resulting in small latency. In contrast, standard instruction-tuned models such as Qwen3-VL-Instruct-8B suffer a dramatic latency surge exceeding 90×. This extreme relative increase underscores the substantial cost of CoT generation. Many closed-source models generally maintain moderate latency growth, which may be attributed to their potential use of implicit reasoning mechanisms that already allocate computational resources. Some medical-specific models demonstrate superior efficiency in terms of correct reasoning steps per unit time, likely due to domain-aligned optimization. LLaVA-Med achieves the highest correct reasoning steps per unit time due to its inherently fast inference speed.

**Impact.** Notably, CoT prompting fails to yield consistent gains in medical image understanding and can even reduce accuracy, likely because it introduces unnecessary or misleading reasoning steps in domains where diagnostic decisions depend more on visual cues than logical inference. The problem is especially pronounced when medical models lack robust multimodal grounding,

and CoT may further raise hallucination risk or distract attention from critical features (Li et al., 2025). Some prior studies have discussed this phenomenon. Mishra & Thakkar (2023) points out that CoT is highly sensitive, and unreasonable reasoning chains may substantially degrade performance. Jiang et al. (2025) measures the effects of CoT in the MME-CoT benchmark: most perception tasks showed decreased performance, while about half of the reasoning tasks declined. Closed-source models exhibit smaller performance drops or even a positive impact under CoT prompting. This suggests that closed-source models possess more robust internal reasoning mechanisms and stronger instruction-following controls, allowing them to absorb the additional CoT constraint without severely disrupting prediction quality. For Qwen3-VL-Thinking, performance changes under CoT prompting suggest that its generated reasoning may be more tightly coupled with the prediction process, where additional structured prompting encourages more consistent step-wise outputs.

**Consistency.** Most models tend to generate similar reasoning steps when handling the same task, resulting in generally high path consistency scores. Most closed-source models achieve relatively high scores, benefiting from consistent generation processes and reasoning patterns. GPT-5 shows the lowest consistency because it often omits intermediate reasoning steps, producing incomplete chains. Open-source models also exhibit relatively high consistency. Aside from the MedGemma series, where larger models outperform smaller ones, models within the same series generally show very similar consistency scores.

## 5.3 Qualitative Analysis

By analyzing model outputs with errors, systematic errors are emerging within the intermediate steps in CoT, rather than merely at the final prediction. For example, an error occurred in the intermediate reasoning steps, where the model misinterpreted secondary features (e.g., narrow anterior chamber angle and iris deposits) as primary evidence, leading it to incorrectly conclude angle-closure glaucoma and deviate from the correct treatment path. Such qualitative inspection highlights three factors:

1. **Incomplete Verification of Decisive Diagnostic Features.** Although the CoT reasoning often identified some relevant abnormalities, it frequently omitted or misweighted critical criteria, such as the extent of epithelial involvement in the pathology case, thereby allowing early misreadings to dominate the conclusion and persist through the subsequent steps.

2. **Weakened Vision-Language Grounding During Step-wise Verbalization.** By forcing the model to translate visual cues into descriptive textual representations before decision-making, CoT increased the risk of information distortion, subtle semantic drift, and gradual loss of fine visual detail. In the hematology example, this intermediate translation process leads to an inaccurate verbal focus on nuclear shape while neglecting the defining cytoplasmic granules, their relative prominence, and characteristic spatial distribution.

3. **Error Accumulation Along the Reasoning Chain.** Once an early descriptive mistake occurred, subsequent steps propagated and rationalized the error, producing a seemingly coherent but ultimately incorrect explanation that became harder to override with additional context.

These observations indicate that the degradation with the CoT prompt reflects deeper vulnerabilities in how visual evidence is interpreted and verified across multiple reasoning stages. Representative examples and detailed error analyses are provided in Appendix D.2.

## 6 Conclusion

In this work, we introduce M3CoTBench, a novel benchmark designed to evaluate CoT reasoning in MLLMs for medical image understanding. Our benchmark addresses the critical gap between answer correctness and reasoning quality in clinical AI systems by incorporating diverse imaging modalities or examination types, step-by-step reasoning annotations, and tailored multi-dimensional evaluation metrics across medical cases of varying difficulty, from simple pattern recognition to complex diagnostic reasoning, enabling fine-grained analysis of model capabilities. Through comprehensive assessments of state-of-the-art MLLMs, we demonstrate limitations of existing models in generating interpretable and clinically aligned reasoning. We hope this benchmark will inspire future research toward more transparent, trustworthy, and practically valuable AI systems for healthcare and beyond. More discussions about limitations and social impact can be seen in Appendix E and Appendix F.

ACKNOWLEDGMENTS AND DISCLOSURE OF FUNDING.

This is A Project Supported by Scientific Research Fund of Zhejiang University (XY2025026). This work is also supported by the National Key R&D Program of China (Grant No. 2025YFF0511302).

ETHICS STATEMENT

We have ensured that our study and dataset construction follow ethical standards, with no direct involvement of human subjects, and no foreseeable risk of harm. Data usage complies with privacy and legal requirements, and we have aimed to mitigate potential biases in annotations and model evaluation. We disclose no conflicts of interest or sponsorship that could influence the results.

REPRODUCIBILITY STATEMENT

We have already elaborated on all the models or algorithms proposed, experimental configurations, and benchmarks used in the experiments in the main body or appendix of this paper. Furthermore, we declare that the entire code used in this work has been released.

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

APPENDIX

This supplementary material provides more detailed information about M3CoTBench. The content of each appendix is summarized as follows:

- **Appendix A.** Provides a detailed description of how large language models are applied in this work. This includes their use in assisting writing, guiding dataset construction, supporting annotation processes, and contributing to model evaluation.

- **Appendix B.** Offers comprehensive information about the dataset used in this study, including the sources of the data, the diseases and abnormalities covered, the distribution of image resolutions, detailed task specifications in the benchmark, and descriptions of CoT annotations.

- **Appendix C.** Provides an in-depth explanation of the evaluation methodology, including the metrics used, the design of prompts, and additional clarifications on how model performance is measured and interpreted.

- **Appendix D.** Presents supplementary experimental results that complement the main paper, along with illustrative case studies that demonstrate model behavior and practical outcomes in various scenarios.

- **Appendix E.** Discusses the known limitations of this study, including potential weaknesses in the methodology, dataset coverage, and model generalizability, providing a balanced view of the research.

- **Appendix F.** Highlights potential societal implications of this work, considering both beneficial applications and possible risks, and reflecting on the broader impact of deploying such models in real-world scenarios.

## A  THE USE OF LARGE LANGUAGE MODELS

We use large language models solely for polishing our writing, and we have conducted a careful check, taking full responsibility for all content in this work. In addition, LLMs and MLLMs were also used in the construction of the dataset and the evaluation of models, and the specific usage has been described in detail in the main text.

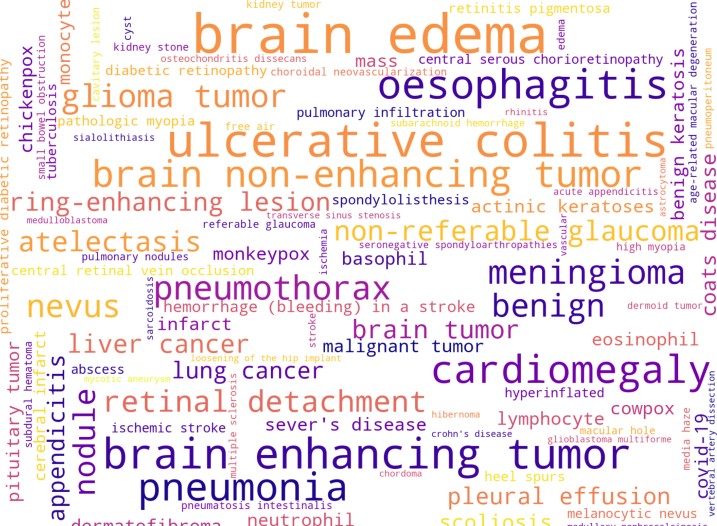

Figure A1: **Word cloud for abnormality and diseases included in M3CoTBench.** The word cloud below visualizes the frequency and variety of these labels, highlighting the spectrum of diagnostic conclusions and imaging findings represented.

# B MORE DETAILS ABOUT THE DATASET

## B.1 SOURCE DATASET INFORMATION

Images in the M3CoTBench dataset are collected from 55 publicly available datasets, offering a highly diverse and representative foundation for training and evaluating multi-modal medical reasoning models. Its comprehensive coverage across modalities, anatomies, time periods, and geographic sources ensures broad applicability and robustness in real-world clinical scenarios. The detailed information of data sources can be seen in Table A1.

## B.2 DISEASES AND ABNORMALITIES

This dataset contains a wide range of diseases and abnormalities. A word cloud illustrating their distribution is shown in Figure A1.

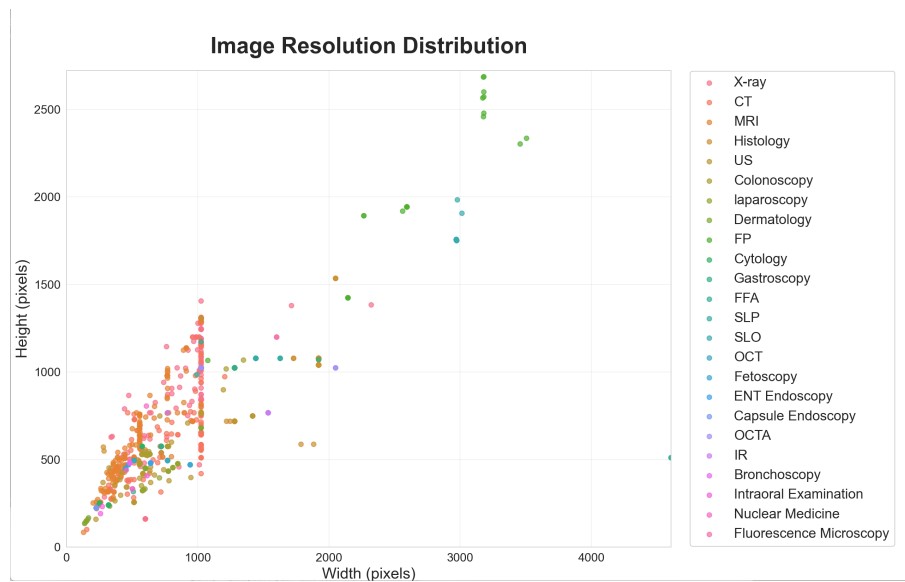

Figure A2: **Image resolution distribution in M3CoTBench.** Most images are concentrated below a width of 1200 and a height of 1500, though some exhibit higher resolutions.

## B.3 IMAGE RESOLUTION DISTRIBUTION

For the images, we retained their original sizes as provided in the source datasets, without applying any additional compression or resizing. Some images may have been preprocessed in their original datasets. However, for tasks such as entity linking, grading, and image quality comparison, we concatenate two images side by side, which results in increased image width. The resolution distribution information can be seen in Figure A2.

- **Diversity in examination types:** The dataset covers 24 imaging modalities and examination methods, which can be grouped into six major categories: ophthalmic imaging, radiology, endoscopy, microscopy, ultrasound-based examinations, and surface-level inspections. These include slit lamp photography (SLP), fundus photography (FP), optical coherence tomography (OCT), optical coherence tomography angiography (OCTA), scanning laser ophthalmoscopy (SLO), fundus fluorescein angiography (FFA), X-ray, computed tomography (CT), magnetic resonance imaging (MRI), ultrasound (US), infrared reflectance (IR), nuclear medicine, fetoscopy, laparoscopy, colonoscopy, gastroscopy, capsule endoscopy, bronchoscopy, ENT endoscopy, histology, cytology, fluorescence microscopy, dermoscopy, and intraoral examination.

- **Diversity in anatomical regions:** The datasets encompass a broad spectrum of anatomical regions, including but not limited to the eye, skin, chest (lungs and heart), brain, abdomen (liver, kidney, stomach, etc.), oral cavity, uterus and fetal environment, breast, vertebrae, hip, knee, foot,

blood, and bone marrow. This anatomical diversity supports the evaluation of models' capability across different clinical tasks and organ systems.

- **Diversity in publication years:** The included datasets were published across a wide temporal range, from earlier benchmarks to very recent contributions. This time span captures the evolution of imaging quality, annotation practices, and diagnostic standards, making the dataset suitable for both historical benchmarking and future-proof model evaluation.

- **Geographic diversity:** The data sources originate from over a dozen countries and regions, reflecting a variety of healthcare environments, population demographics, and medical imaging protocols. This geographic diversity enhances the robustness, fairness, and real-world applicability of models trained on the dataset, particularly in cross-domain or multi-institutional settings. The geographic distribution of data sources is illustrated in FigureA3.

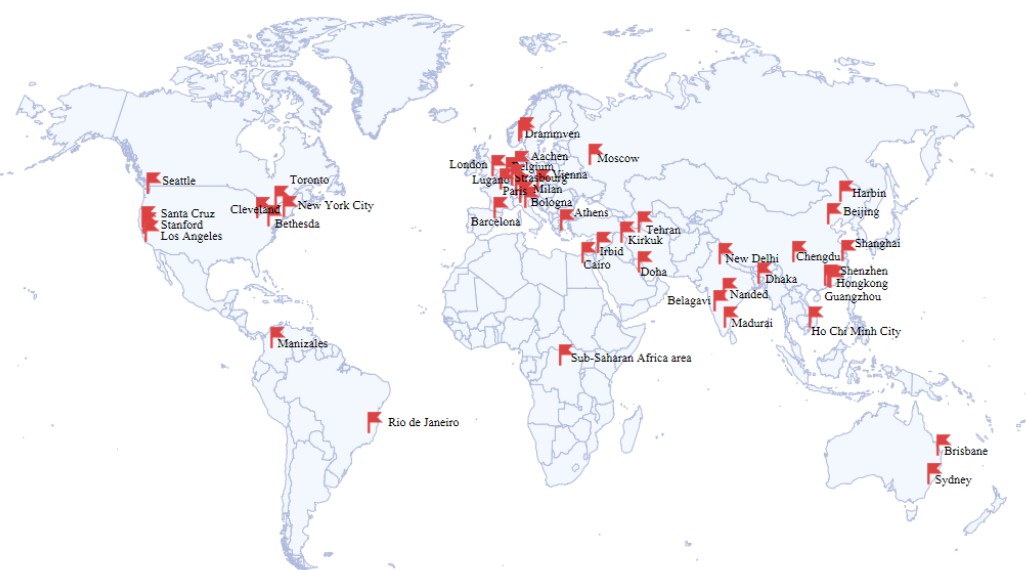

Figure A3: Geographic distribution of data sources in the dataset. Red flags indicate the locations of contributing hospitals or institutions, where applicable. Due to the complex and varied origins of some datasets, exact source locations may not always be clearly identifiable.

## B.4 DETAILED INTRODUCTION TO TASKS

The benchmark encompasses a diverse range of tasks that mirror real-world clinical challenges in medical visual-language reasoning. These tasks are designed to evaluate not only a model's ability to recognize and classify visual information, but also its capacity to comprehend spatial, procedural, and diagnostic contexts. Broadly, the tasks can be grouped into two conceptual levels: **Perceptual-level tasks** focus on low- to mid-level visual understanding, such as identifying image modality, recognizing anatomical structures, or assessing image quality. These tasks primarily test the model's capability to extract and interpret observable features from the image. **Knowledge-based reasoning tasks**, on the other hand, require integration of visual features with clinical knowledge, commonsense reasoning, or multi-step inference. These include complex tasks such as diagnosing diseases, predicting disease progression, grading severity, planning clinical actions, or identifying causal relationships.

- **Modality / Examination Types:** Understanding and recognizing the imaging modality involved, such as CT, MRI, X-ray, or OCT, demonstrates the model's awareness of different diagnostic techniques and their clinical contexts.

- **Image Quality Assessment:** Evaluating whether an image is diagnostically adequate, and comparing the relative quality between multiple images when necessary. This reflects the model's ability to judge image usability in clinical practice.

Table A1: Data sources of different modalities in M3CoTBench

| Dataset | Anatomical Region | Modality / Examination Type |
|---|---|---|
| OphthalVQA (Xu et al., 2023) | Eye | SLP, FP, OCT, US, SLO, FFA |
| IDRiD (Porwal et al., 2018) | Eye | FP |
| JustRAIGS (Rotterdam Ophthalmic Institute, 2024) | Eye | FP |
| RIADD (Quellec et al., 2020) | Eye | FP |
| DRAC 2022 (Qian et al., 2023) | Eye | OCTA |
| RAVIR (Hatamizadeh et al., 2022; Hatamizadeh, 2020) | Eye | IR |
| ISIC 2018 (Codella et al., 2019) | Skin | Dermoscopy |
| HAM10000 (Tschandl et al., 2018) | Skin | Dermoscopy |
| MSLD v2.0 (Ali et al., 2024) | Skin | Dermoscopy |
| VQA-RAD (Lau et al., 2018) | Chest, Abdomen, Brain | X-ray, CT, MRI, Nuclear Medicine |
| VQA-Med-2019 (Ben Abacha et al., 2019) | Chest, Abdomen, Brain | X-ray, CT, MRI, US |
| SLAKE (Liu et al., 2021) | Chest, Abdomen, Brain | X-ray, CT, MRI |
| Chest XR COVID-19 (Akhloufi & Chetoui, 2021) | Chest (Lung) | X-ray |
| TB Chest X-ray (Rahman et al., 2020) | Chest (Lung) | X-ray |
| Heel Bone (Taher & Özacar, 2024) | Foot | X-ray |
| Digital Knee X-ray (Gornale & Patravali, 2020) | Knee | X-ray |
| Vertebrae X-ray (Fraiwan et al., 2022) | Vertebrae | X-ray |
| Hip Implant X-ray (Rahman et al., 2022) | Shoulder | X-ray |
| CT Kidney (Islam et al., 2022) | Kidney | CT |
| COVID-19 Lung CT (Zhao et al., 2020) | Lung | CT |
| Brain Stroke CT (Koç et al., 2022) | Brain | CT |
| LDCTIQAC 2023 (Lee et al., 2025) | Abdomen | CT |
| MSCT-Image Dataset (Sharifullin et al., 2020) | Brain | CT |
| PENGWIN (Liu et al., 2025a;b) | Pelvis | CT |
| ToothFairy (Bolelli et al., 2024; Lumetti et al., 2024; Cipriano et al., 2022) | Oral Cavity | CT |
| Brain Tumor (Bhuvaji et al., 2020) | Brain | MRI |
| LGG Segmentation (Buda et al., 2019) | Brain | MRI |
| Brain Cancer MRI (Rahman, 2024) | Brain | MRI |
| BRATS-SSA (Adewole et al., 2023) | Brain | MRI |
| Cancer-Net PCa-Data (Wong et al., 2022; Gunraj et al., 2023) | Prostate | MRI |
| SIMON MRI (Duchesne et al., 2019) | Brain | MRI |

Table A1 (continued): Data sources of different modalities in M3CoTBench

| Dataset | Anatomical Region | Modality / Examination Type |
|---|---|---|
| BUSI (Al-Dhabyani et al., 2020) | Breast | US |
| FH-PS-AOP (Jieyun & Zhan-Hong, 2023) | Fetal | US |
| Nerve Segmentation (Montoya et al., 2016) | Neck | US |
| Carotid Artery (Momot, 2022) | Neck | US |
| Liver-US (She et al., 2025) | Liver | US |
| Annotated Liver US Dataset (Yiming et al., 2022) | Liver | US |
| BUS-BRA (Gómez-Flores et al., 2024) | Breast | US |
| Quilt-VQA (Seyfioglu et al., 2024) | Multi-regions | Histology |
| BCI (Liu et al., 2022) | Breast | Histology |
| Colorectal Histology MNIST (Kather et al., 2016) | Colon and Rectum | Histology |
| GCHTID (Lou et al., 2024) | Stomach | Histology |
| BACH (Aresta et al., 2019) | Breast | Histology |
| CMIA Histological Slides | Lung, Breast | Histology |
| Fluorescent Neuronal Cells (Morelli et al., 2021) | Brain | Fluorescent Microscopy |
| BCCD (shenggan et al., 2018) | Blood | Cytology |
| Raabin-WBC (Kouzehkanan et al., 2022) | Blood | Cytology |
| BMC (Matek et al., 2021a;b) | Bone Marrow | Cytology |
| EndoVis-17-VLQA (Allan et al., 2019) | Abdomen | Laparoscopy |
| m2cai16-tool (Twinanda et al., 2016) | Abdomen | Laparoscopy |
| ImageCLEFmed MEDVQA-GI (Hicks et al., 2023) | Gastrointestinal Tract | Colonoscopy, Gastroscopy |
| Bronchoscopy Dataset (Deng et al., 2023) | Airway Tract | Bronchoscopy |
| Capsule Vision 2024 (Handa et al., 2025) | Gastrointestinal Tract | Capsule Endoscopy |
| ENTRep Challenge 2025 (ENT, 2025) | Ear, Nose, Throat | ENT Endoscopy |
| FetReg (Bano et al., 2021) | Uterus / Fetal Environment | Fetoscopy |
| Fetoscopy Placenta Data (Bano et al., 2020) | Uterus / Fetal Environment | Fetoscopy |
| Dental Condition Dataset (Sajid, 2024) | Oral Cavity | Intraoral Examination |

- **Recognition:** General visual recognition tasks, including identifying anatomical structures, tissues, or medical devices, without explicit spatial reference.

- **Referring Recognition:** Region-specific identification tasks where the model must recognize or interpret a particular area in the image based on the question or accompanying text.

- **Counting:** Quantifying specific elements in an image, such as surgical tools, lesions, polyps, or cells, often requiring precise object detection and differentiation.

- **Localization:** Identifying the spatial location of regions of interest, such as lesions, organs, or abnormal structures, testing the model's understanding of spatial relations and context.

- **Diagnosis:** Inferring the presence of abnormalities, diseases, or clinical conditions based on image and text input; this is the most common and clinically important task category.

- **Grading:** Assessing the severity or stage of a medical condition, such as cancer staging or diabetic retinopathy levels, requires a nuanced interpretation of visual cues.

- **Symptom Identification:** Recognizing observable clinical signs or inferring underlying symptoms based on the visual features of the image and contextual cues.

- **Clinical Action Planning:** Making decisions about the next steps in patient care, such as recommending further examinations, procedures, or treatment options, demonstrating clinical reasoning ability.

- **Prediction:** Estimating future disease progression, risks of complications, or expected outcomes, often involving multi-modal reasoning over image and text inputs.

- **Functional Understanding:** Interpreting the physiological function of organs, the intended use of medical instruments, or the purpose of surgical actions, integrating procedural and anatomical knowledge.

- **Causal Reasoning:** Identifying the cause or etiology of a symptom or condition, requiring the model to reason about potential underlying mechanisms or prior events.

## B.5 CoT Annotation

The CoT annotations are collaboratively generated by medical experts and MLLMs, generally following a four-part structure: {examination type, key feature, key conclusion, additional analysis}. This approach aligns closely with clinical reasoning patterns used by physicians, who often begin by identifying the type of examination or modality, observing key findings, deriving conclusions, and, when necessary, conducting further interpretation or differential diagnosis. The length and structure of CoT vary depending on the task. For tasks such as recognition, diagnosis, and grading, a three-step format,{examination type, key feature, key conclusion}, is generally sufficient. In contrast, more complex tasks like treatment planning, causal reasoning, symptom analysis, prognostic prediction, or functional interpretation often require a four-step annotation to capture the depth of reasoning. When it comes to identifying the imaging modality, CoT length depends on the nature of the question. For example, in general tasks, it may not be necessary to analyze image features to identify the modality explicitly. However, in questions specifically targeting modality identification, CoT annotations typically include two steps, focusing on characteristic visual clues about the imaging technique used. Notably, during examination modality statistics, some subtypes are grouped into broader categories. However, in CoT annotations, these modalities are often distinguished more finely. For example, IHC and HE are treated separately, as are MRI T1-weighted and T2-weighted images. Examples of CoT annotation are shown in Figure A4 and Figure A5.

## C  Supplementary to Evaluation Processes

### C.1  Examples of Path Similarity in Reasoning Consistency Evaluation

To evaluate the structural stability of reasoning in multi-step tasks, we introduce a **path consistency** metric that measures the similarity of reasoning paths across instances of the same task type. Unlike traditional metrics that treat reasoning steps as unordered, this metric accounts for the sequential structure by comparing generated paths to a canonical reference using the normalized longest common subsequence (LCS). The final score, averaged over all 13 tasks, reflects the model's ability to follow

**a)** What examinations need to be done next?

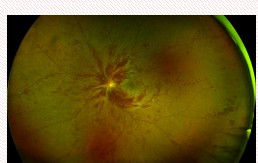

**CoT Annotation**

1. Examination type: Scanning laser ophthalmoscopy (SLO).

2. Key features: Marked tortuosity and dilation of retinal veins, extensive blot and flame-shaped hemorrhages, especially around the optic disc.

3. Key conclusion: The likely diagnosis based on this image is central retinal vein occlusion.

4. Additional Analysis (action suggestions): Fluorescein angiography demonstrates retinal circulation patterns and identifies areas of nonperfusion, while OCT reveals macular edema and structural changes characteristic of venous occlusive disease.

**b)** What might be the cause of this condition?

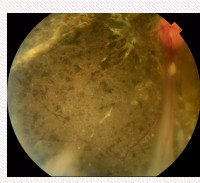

**CoT Annotation**

1. Examination type: Fundus photography.

2. Key features: The image shows retinal folds and distortion around the optic disc with scattered pigmentary deposits.

3. Key conclusion: The likely diagnosis based on this image could be suspicious chronic tractional retinal detachment(TRD).

4. Additional Analysis (causal Reasoning): Chronic TRD develops from prolonged fibrovascular proliferation creating mechanical traction on retinal tissue, commonly seen in advanced proliferative diabetic retinopathy with inadequate glycemic control. .

**c)** What disease is most likely associated with the picture? (Select one option)
A. Liver Cancer  B. Hepatitis
C. Cirrhosis      D. Fatty liver disease

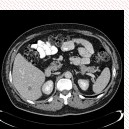

**CoT Annotation**

1. Examination type: CT.

2. Key features: The liver contains an irregular, hypodense mass with heterogeneous enhancement.

3. Key conclusion: The likely diagnosis is liver cancer.

**d)** What lifestyle measures should be taken to manage this situation?
(Select all that apply)
A. Reduce salt intake
B. Engage in regular moderate exercise
C. Smoke more to relieve stress
D. Avoid excessive alcohol consumption
E. Maintain a healthy weight
F. Skip medications without consulting a doctor
G. Monitor blood pressure regularly
H. Eat more processed foods

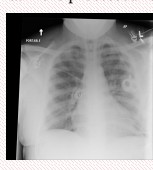

**CoT Annotation**

1. Examination type: X-ray.

2. Key features: The cardiac silhouette is enlarged, occupying more than half of the thoracic cavity's transverse diameter.

3. Key conclusion: The likely diagnosis is cardiomegaly.

4.Additional Analysis (action suggestions/option analysis): Option A: Salt reduction prevents fluid retention and hypertension. Option B: Moderate exercise strengthens cardiovascular system under guidance. Option C: Smoking worsens cardiovascular disease directly. Option D: Alcohol excess causes cardiomyopathy and arrhythmias. Option E: Weight loss reduces cardiac workload. Option F: Medication adherence crucial for disease management. Option G: Blood pressure monitoring ensures treatment effectiveness. Option H: Processed foods worsen hypertension and inflammation.

**e)** What surgical or medical instruments visible in the image? (Select one option)
A. No instruments present
B. Scalpel
C. Metal clip
D. Surgical sponge

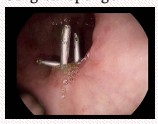

**CoT Annotation**

1. Examination type: Colonoscopy.

2. Key features: Three separate silver-colored cylindrical structures in the image.

3. Key conclusion: There are three metal clips in the image.

Figure A4: **Examples of CoT annotations with corresponding images and questions in M3CoTBench (1).** Different types of questions are annotated with different lengths of CoT steps. For example, diagnostic **(c)** and recognition **(e)** questions involve three annotation steps, while action-planning **(a, d)** and causal analysis **(b)** questions are annotated with four steps.

**a)** True or False: This abnormality will certainly progress to squamous cell carcinoma if untreated.

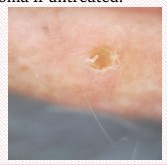

**CoT Annotation**

1. Examination type: Dermatological image.

2. Key features: Localized rough, scaly patch with a central depression, yellowish adherent scales, and surrounding erythema.

3. Key conclusion: The likely diagnosis is actinic keratoses.

4. Additional Analysis (prediction): Only a small percentage (estimated around 0.1–10% per lesion per year) may evolve into squamous cell carcinoma.

**b)** How many polyps are in the image?

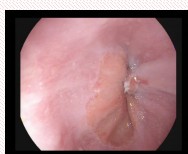

**CoT Annotation**

1. Examination type: Gastroscopy.

2. Key features: The mucosal surface appears smooth and uniform with natural folds but lacks any visible elevated or protuberant masses..

3. Key conclusion: There is no polyp in the image.

**c)** True or False: The left image shows higher knee osteoarthritis severity than the right.

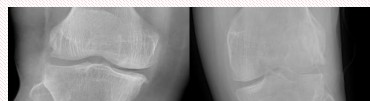

**CoT Annotation**

1. Examination type: X-ray.

2. Key features: In the left image, joint space is relatively well preserved; only mild narrowing. In the right image, obvious joint space narrowing, subchondral sclerosis, and osteophyte formation.

3. Key conclusion: The right image shows higher knee osteoarthritis severity than the left.

**d)** The figure consists of two images side by side. The image on the right is a segmentation mask of a specific region in the image on the left. What does the white area in the right image represent in the left image? (Select one option)
A. Glioma lesion  B. Brain edema
C. Cerebrospinal fluid  D. Skull
E. Normal brain tissue

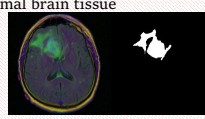

**CoT Annotation**

1. Examination type: MRI(FLAIR).

2. Key features: The right image shows an irregular white area with increased signal intensity, corresponding to the bright hyperintense region in the left cranial image.

3. Key conclusion: The white area in the right image likely represents glioma tumor.

**e)** Which of the following symptoms is this patient most likely to experience?  (Select all that apply)

A) RUQ pain (Right Upper Quadrant pain)
B) Jaundice  C) Dark urine
D) Clay-colored stools  E) Weight loss

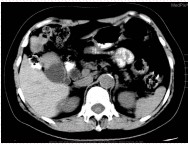

**CoT Annotation**

1. Examination type: CT.

2. Key features: Enlarged gallbladder with low-attenuation lumen and dilated common bile duct alongside dilated intrahepatic bile ducts.

3. Key conclusion: The possible diagnosis is cystic duct and CBD obstruction.

4. Additional Analysis (symptom analysis): Obstruction of the cystic duct and common bile duct (CBD) can lead to bile stasis, causing: RUQ pain due to gallbladder distension or inflammation. Jaundice from buildup of bilirubin. Dark urine because conjugated bilirubin is excreted in urine. Clay-colored stools due to lack of bile pigments in the intestines.

Figure A5: **Examples of CoT annotations with corresponding images and questions in M3CoTBench (3).** Different types of questions are annotated with different lengths of CoT steps. For example, counting **(b)**, grading **(c)** and referring recognition **(d)** questions involve three annotation steps, and prediction **(a)** and symptom **(e)** questions are annotated with four steps.

consistent, interpretable reasoning patterns, a key property for transparency and trust in medical decision-making. Here are some examples to show the specific calculation method:

Consider the following reasoning paths, where each element is one of {*modality*, *feature*, *diagnosis*, *treatment*}, representing a progression from identifying the imaging type, describing visual findings, inferring clinical conditions, to suggesting appropriate medical interventions.

- **Example 1:** $P_1 = $ [modality, feature, diagnosis], $P_2 = $ [feature, modality, diagnosis]. Then the LCS is [modality, diagnosis] and [feature, diagnosis]. $|\text{LCS}(P_1, P_2)| = 2$, thus

$$\text{sim}(P_1, P_2) \;=\; \frac{2}{\max(3,3)} \;=\; \tfrac{2}{3} \approx 0.67. \tag{A1}$$

- **Example 2:** $P_1 = $ [modality, diagnosis, treatment], $P_2 = $ [modality, feature, diagnosis, treatment]. Then the LCS is [modality, diagnosis, treatment] $|\text{LCS}(P_1, P_2)| = 3$, thus

$$\text{sim}(P_1, P_2) \;=\; \frac{3}{\max(3,4)} \;=\; \tfrac{3}{4} = 0.75. \tag{A2}$$

- **Example 3:** $P_1 = $ [modality, feature, treatment] , $P_2 = $ [modality, feature, diagnosis, treatment]. Then the LCS is [modality, feature, treatment] The $|\text{LCS}(P_1, P_2)| = 3$, thus

$$\text{sim}(P_1, P_2) \;=\; \frac{3}{\max(3,4)} \;=\; \tfrac{3}{4} = 0.75. \tag{A3}$$

- **Example 4:**

  $P_1 = $ [feature, modality, diagnosis, treatment] , $P_2 = $ [modality, feature, diagnosis, treatment]. Then the LCS is [modality, diagnosis, treatment] and [feature, diagnosis, treatment] The $|\text{LCS}(P_1, P_2)| = 3$, thus

$$\text{sim}(P_1, P_2) \;=\; \frac{3}{\max(3,4)} \;=\; \tfrac{3}{4} = 0.75. \tag{A4}$$

## C.2 EVALUATION PROMPTS

During evaluation, we use GPT-4o and LLaMA-3.3-70B-Instruct-Turbo to assess the accuracy of answers to the questions. Accuracies of the direct answers and CoT answers are both averaged from the two models. We also use GPT-4o to assess the correctness of each step, and use GPT-4o and Gemini 2.5 to determine the step order in the CoT output. The consistency scores of the steps in CoT outputs are averaged from the two models. Since the feature description and additional analysis parts are relatively subjective, with multiple valid expressions for the same meaning, we adopt more lenient instructions for these components. In contrast, we apply stricter criteria to the examination modality and key conclusion steps.

### C.2.1 EVALUATION PROMPTS FOR ANSWER ACCURACY

The prompt for calculating accuracy for both direct outputs and CoT outputs is shown below:

---

**Prompt for calculating accuracy for both direct outputs and CoT outputs**

**You are a medical evaluation expert:**

#Your tasks:
1. Extract the final answer only from the model's prediction below. 2. Judge if it matches the provided ground-truth answer.
#Type instruction:
Return ONLY a JSON object with the EXACT format below (no extra text):

```
[
{{
  "match": true or false,
  "final_answer": "the extracted final
  answer text"
}}
Inputs:

Question:
{question}

Ground-truth Answer:
{answer}

Model's Prediction:
{prediction}
```

---

### C.2.2 EVALUATION PROMPTS FOR PRECISION CALCULATION

The prompt for precision calculation is:

---

**Prompt for calculating precision for CoT outputs**

**Given a solution with multiple reasoning steps for an image-based problem, reformat it into well-structured steps and evaluate their correctness.**

Step 1: Reformatting the Solution
Convert the unstructured solution into distinct reasoning steps while:

- Preserving all original content and order.
- Not adding new interpretations.
- Not omitting any steps.

# Step Types

1. Image Modality or Examination Types
   - Describes the imaging type or procedure used (e.g., CT, MRI, histology, endoscopy, Anterior segment slit lamp examination, scanning laser ophthalmoscopy).
   - Focuses on technical aspects without interpretation.

2. Key Image Feature Analysis
   - Pure visual observations obtained from the image.
   - Describes visible structures or abnormalities in the image. (e.g., The eye exhibits notable redness).
   - Pure observation without inference.

3. Identification, Localization, or Diagnostic Conclusions or other Conclusions
   - Provides specific findings or diagnosis based on image features. (e.g., the likely diagnosis based on this image could be Coats' disease).

---

- Includes reasoning and clinical conclusions.
- T The location of the abnormalities or organs.
- Classification conclusion for cells or organs. (e.g., The cell is a Basophil.)
- Recognition conclusion for instruments or their processes. (e.g., The current state of the monopolar curved scissors is cauterization.)
- Other conclusions. (e.g., The red area in the right image represents in the left image is the pubic symphysis.)

4. Knowledge-Based / Differential / Exploratory Analysis

- This is not mandatory and is only required for those requiring additional analysis.
- Includes disease progression prediction, organ/cell function, treatment or further examination suggestions, cause analysis of disease or abnormalities, other medical knowledge, and step-by-step analysis of multiple-choice options.

# Step Requirements

- Each step must be atomic (one conclusion per step)
- No content duplication across steps

Step 2: Evaluating Correctness
Evaluate each step against:
# Ground Truth Matching
For modality or examination types:

- Must strictly correspond to ground truth; different wording allowed if meaning is equivalent.
- MRI T1/T2/DWI sequences are considered different modalities. Endoscopy answers must exactly match the type, e.g., ENT Endoscopy, gastroscopy, capsule endoscopy, laparoscopy and so on.
- Various synonymous expressions for 'Section stained with hematoxylin and eosin (H&E)', e.g., H&E-stained slide are all acceptable.

For image feature description:

- Mostly-overlap matching: Answers that match the majority of the ground truth and convey the same meaning are considered correct.

For key conclusions:

- Should strictly correspond to ground truth; different wording allowed if meaning is equivalent.

For additional analysis:

- Mostly-overlap matching: Answers that match the majority of the ground truth and convey the same meaning are considered correct.

# Reasonableness Check

- Logic is valid
- Step information must not contradict any ground truth or correct answer
- Step information must support or be neutral to correct answer

# Judgement Categories

- Match: Aligns with ground truth.
- Wrong: Invalid or contradictory.
- N/A: For background information steps.

# Output Requirements

1. The output format MUST be in valid `JSON` format without ANY other content.
2. For highly repetitive patterns, output it as a single step.

Here is the JSON output format:

```
[
{
    "step_type": "modality or examination types
    |image feature description|key conclusions
    |additional analysis
    |Restatement of the question."
    "Step information": "Step result",
    "judgment": "Match|Wrong|N/A"
  }
]
```

Your task is to reformat the following solution into discrete reasoning steps and evaluate each step based on the ground truth.
Input:

```
[Problem]

{question}

[Solution]

{solution}

[Correct Answer]

{answer}

[Ground Truth Information]

{gt\_annotation}
```

### C.2.3 EVALUATION PROMPTS FOR RECALL CALCULATION

The prompt for recall calculation is:

---
**Prompt for calculating recall for CoT outputs**

**You are an expert system for verifying solutions to medical image-based problems. Your task is to match the ground truth middle steps with the provided solution.**

# Input Format:
1. Problem: The original question/task.
2. A Solution of a model.
3. Ground Truth: Essential steps required for a correct answer.

# Matching Process:
You need to match each ground truth middle step with the solution. Match Criteria:

- The middle step should match the content or be directly entailed by a certain content in the solution.

- For subjective or descriptive steps such as image feature descriptions, treatment suggestions, disease causes, symptoms or cellular/instruments functions, match leniently: A step is "Matched" if the overall meaning largely overlaps with the solution and there is no contradiction, even if wording differs. Exact wording or structure is not required as long as the clinical implication is preserved.

- For objective steps such as specific diseases, cell names, lesion names, or image modalities/examination types, match strictly: The terminology must refer to the same medical

---

concept, though phrasing may differ (e.g., "retinal detachment" vs. "detached retina" is acceptable).

- For modality or examination types, it must strictly correspond to ground truth; different wording is allowed if the meaning is equivalent. Different types of MRI sequences, like T1/T2/DWI are considered different modalities. Endoscopy answers must exactly match the type, e.g., ENT Endoscopy, gastroscopy, capsule endoscopy, laparoscopy, and so on. Various synonymous expressions for 'Section stained with hematoxylin and eosin (H&E)', e.g., H&E-stained slide are all acceptable.

In all cases, evaluate whether each ground truth step is represented in the solution, either explicitly or with clear implication.

# Output Format:

JSON array of judgments:

```
[
  {
    "step_index": <integer>,
    "step_type": "modality or examination types
    |image feature description
    |key conclusions|additional analysis
    |restatement of the question"
    "judgment": "Matched" | "Unmatched",
  }
]
```

# Additional Rules:

1. Only output the JSON array with no additional information.

2. Judge each ground truth middle step in order, without omitting any step.

Here is the problem, answer, solution, and the ground truth middle steps:

```
[Problem]

{question}

[Answer]

{answer}

[Solution]

{solution}

[Ground Truth Information]

{gt_annotation}
```

### C.2.4 EVALUATION PROMPTS FOR STEP ORDER RECOGNITION

When computing CoT consistency, it is necessary to determine the order of the reasoning steps in the model's output. This requires first classifying the type of each step. Our prompt is as follows:

---

**Prompt for step order recognition**

**Example JSON output structure:**

```
{
  "modality_order": 1,
  "feature_order": 2,
```

```
    "conclusion_order": 3,
    "others_order": 4,
    "modality_subs": ["substring 1", "substring 2"],
    "feature_subs": ["substring 1", "substring 2"],
    "conclusion_subs": ["substring 1", "substring 2"],
    "others_subs": ["substring 1", "substring 2"]
}
```

**System Prompt:** You are a medical reasoning pathway analyzer. Analyze an AI's answer to a medical question by extracting information into four categories and determining their appearance order.

**Medical Question/Task:** Not provided

**Categories:**

- **1. modality_subs** - Imaging/examination methods.
  Extract the image modality or examination type.
  Examples: "Fundus photography", "MRI (FLAIR)", "Anterior segment slit lamp examination", "CT scan", "Ultrasound B-scan", "Section stained with hematoxylin and eosin (H&E)", "microscopy", "ENT Endoscopy"

- **2. feature_subs** - Characteristics visible directly from the image.
  Examples: "opaque lens with dense white structure", "ill-defined hyperintense lesion with heterogeneous internal signal in the right cerebral hemisphere", "mild sulcal effacement, no significant mass effect", "cross-sectional musculoskeletal structures with visible bone contours", "bright, dense structure visible in the right kidney's collecting system"

- **3. conclusion_subs** - Key conclusions including diagnoses, recognition (cell/organ/instrument/types/surgical processes), anatomical locations, grading, counting.
  Examples: "mature cataract", "consistent with low grade diffuse astrocytoma", "kidney stone", "left kidney", "Coats' disease", "Monocyte cell type", "The instrument in the image is a Tube", "The state of ultrasound probe is idle"

- **4. others_subs** - Further explanations, treatment info, action recommendations, symptoms, functions, or clinical knowledge.
  Examples: "Visual function assessment is needed", "Dehydration concentrates urine, increasing supersaturation of stone-forming salts", "COVID-19 is caused by the SARS-CoV-2 virus", "For benign lesions, aggressive treatments like chemotherapy are usually unnecessary", "Option A: Dehydration increases risk. Option B: High oxalate intake promotes stones"

**Extraction Rules:**

- Extract complete phrases or sentences for each category.

- Extract multiple distinct substrings per category if present.

- Copy exact text; do not paraphrase.

- Use empty array [] if no content.

**Order Rules:**

- Scan text from start to end to determine which category appears first, second, third, fourth.

- Assign 1-4 based on appearance sequence.

- Assign 0 if category does not appear.

- Examples: (1,2,3,4) = modality → feature → conclusion → others, (2,3,1,0) = feature → conclusion → modality, no others.

**Example Input:** "Fundus photography of the right eye shows an opaque lens with dense white structure. This appearance is consistent with mature cataract. Visual function assessment is needed."

**Example Output:**

```
{
    "modality_order": 1,
```

```
    "feature_order": 2,
    "conclusion_order": 3,
    "others_order": 4,
    "modality_subs": ["Fundus photography"],
    "feature_subs": ["opaque lens with dense white structure"],
    "conclusion_subs": ["mature cataract"],
    "others_subs": ["Visual function assessment is needed"]
}
```
**CRITICAL:** You must respond with ONLY valid JSON format. Do not include any other text before or after the JSON object.
**Your output must be valid JSON in this exact format: {OUTPUT_FORMAT}**

## D   SUPPLEMENTARY TO EXPERIMENTS

### D.1   SUPPLEMENTARY RESULTS

Here, we present the average response time per question for each MLLM under both the direct and step-by-step settings, as shown in the Table A2. From the table, it is evident that CoT reasoning generally increases the average response time compared to direct output, as generating a full sequence of reasoning steps requires more computation and context processing. Overall, response time depends not only on the generation mode (direct vs. CoT) but also on model size, architecture, and built-in reasoning mechanisms.

Table A2: Comparison of the average response time per question for MLLMs under direct and step-by-step reasoning conditions. Optimal / sub-optimal results are highlighted in **bold** / underline.

| Model | $T_{\text{direct}}$(s/sample) | $T_{\text{CoT}}$(s/sample) |
|---|---|---|
| LLava-CoT Xu et al. (2025) | 0.70 | 7.78 |
| InternVL3.5-8B (Wang et al., 2025a) | 0.93 | 16.91 |
| InternVL3.5-30B (Wang et al., 2025a) | 4.01 | 66.92 |
| Qwen3-VL-Instruct-8B (Bai et al., 2025) | **0.52** | 49.10 |
| Qwen3-VL-Instruct-30B (Bai et al., 2025) | 1.69 | 60.34 |
| Qwen3-VL-Thinking-8B (Bai et al., 2025) | 34.98 | 97.56 |
| Qwen3-VL-Thinking-30B (Bai et al., 2025) | 104.03 | 120.08 |
| GPT-4.1 (OpenAI, 2025a) | 2.37 | 12.04 |
| GPT-5 (OpenAI, 2025b) | 24.26 | 26.58 |
| Gemini 2.5 Pro (Google DeepMind, 2024) | 14.24 | 21.62 |
| Claude-Sonnet-4.5 (Anthropic, 2025) | 4.68 | 12.60 |
| LLaVA-Med (7B) (Li et al., 2023) | 0.77 | **2.49** |
| HuatuoGPT-Vision (7B) (Chen et al., 2024a) | 1.22 | 7.20 |
| HealthGPT (3.8B) (Lin et al., 2025) | 0.92 | 14.07 |
| Lingshu-7B (Team et al., 2025) | 0.66 | 5.56 |
| Lingshu-32B (Team et al., 2025) | 0.78 | 8.44 |
| MedGemma-4B (Sellergren et al., 2025) | 1.49 | 30.76 |
| MedGemma-27B (Sellergren et al., 2025) | 2.58 | 61.06 |

### D.2   CASE STUDY

#### D.2.1   EXAMPLE 1

Comparison of answers from Qwen3-VL-Instruct-30B and the annotated CoT steps.

Q: True or False: The cell shown in the image is Lymphocyte.

Correct Answer: False.

The direct answer is False, and the CoT Answer is True.

**The annotated CoT steps are shown below:** *1. Hematology or Cytology*

*2. The cell has a prominent bilobed nucleus with abundant pinkish-orange granules in the cytoplasm.*

*3. The cell type is Eosinophil.*

*To determine whether the cell shown in the image is a lymphocyte, let's analyze the key features of a lymphocyte and compare them to the cell in the image.*

*Step 1: Identify the cell in the image*

*- The image shows a cell with a large, round nucleus that occupies most of the cell's volume.*

*- The cytoplasm is scant and appears light blue or pale.*

*- The nucleus is dense and dark purple, with a smooth, round contour.*

*- There is no visible granulation in the cytoplasm.*

*Step 2: Recall the characteristics of a lymphocyte*

*- Lymphocytes are a type of white blood cell.*

*- They are typically small to medium in size.*

*- The nucleus is large, round, and dense, often taking up most of the cell's volume.*

*- The cytoplasm is minimal and pale, often appearing as a thin rim around the nucleus.*

*- Lymphocytes do not have granules in their cytoplasm.*

*- The nucleus is usually darkly stained (basophilic), indicating high RNA content.*

*Step 3: Compare the observed cell to lymphocyte characteristics*

*- The cell in the image has a large, round, darkly stained nucleus.*

*- The cytoplasm is minimal and pale.*

*- There are no visible granules.*

*- These features are consistent with a lymphocyte.*

*Step 4: Consider other possibilities*

*- Neutrophils: Have a multi-lobed nucleus and granular cytoplasm.*

*- Eosinophils: Have a bilobed nucleus and large, eosinophilic granules.*

*- Basophils: Have a lobed nucleus and large, basophilic granules.*

*- Monocytes: Have a large, kidney-shaped or horseshoe-shaped nucleus and abundant cytoplasm.*

*- The cell in the image does not match any of these.*

*Step 5: Conclusion*

*- The cell in the image exhibits the classic features of a lymphocyte.*

*Final Answer: True"*

**Analysis**: Introducing CoT reasoning in this case led to worse performance because it caused the model to misweight and misinterpret key diagnostic features. Instead of focusing on the defining characteristics of an eosinophil, namely the bilobed nucleus and abundant pink-orange cytoplasmic granules, the CoT reasoning overemphasized general patterns like a large, round nucleus and scant cytoplasm, which are typical of lymphocytes. The multi-step natural language explanation allowed the model to generate a seemingly coherent rationale, but this reasoning was not properly grounded in the observable image features, creating a self-consistent but ultimately incorrect inference. This demonstrates that CoT can inadvertently introduce cognitive bias and distract attention from critical domain-specific visual cues, leading to errors that direct answer prediction would have avoided.

### D.2.2 EXAMPLE 2

Comparison of answers from MedGemma-27B and the annotated CoT steps.

What might be the treatment options for this condition? (Select one option)

A) Surgery as the first line of treatment

B) Topical corticosteroid drops and dilating drops, with systemic medication when necessary

C) No treatment is required

D) Laser therapy

Correct Answer: B.

The direct answer is B, and the CoT Answer is D.

**The annotated CoT steps are shown below:** *1. Anterior segment slit lamp examination.*

*2. Posterior synechiae causing pupil deformation and keratic precipitates on the corneal endothelium.*

*3. The likely diagnosis based on this image is anterior uveitis.*

*4. A) Surgery as the first line of treatment: Not appropriate for anterior uveitis, as surgery is rarely needed unless complications occur. B) Topical corticosteroid drops and dilating drops, with systemic medication when necessary: This is the standard treatment for anterior uveitis. It helps reduce inflammation and prevent complications such as synechiae. C) No treatment is required: Incorrect, because untreated anterior uveitis can lead to serious complications and vision loss. D) Laser therapy: Not typically used for uveitis. It is more relevant for other eye conditions such as glaucoma or retinal diseases.*

**The output of CoT prediction is shown below:** *The user wants me to identify the condition shown in the image and then select the appropriate treatment option from the given choices.*

*1. **Analyze the image:** The image shows a close-up view of an eye. The iris appears to be pulled forward, and the anterior chamber angle (the space between the iris and the cornea) seems narrow. There is also a white, slightly raised lesion on the iris, which could be a deposit or a sign of inflammation. The pupil is somewhat constricted. The overall appearance suggests a potential issue with the anterior chamber angle and possibly inflammation or pigment dispersion.*

*2. **Identify the likely condition:** Based on the narrow angle appearance, the condition is likely **angle-closure glaucoma** or a related condition like **pigmentary glaucoma** or **pseudoexfoliation glaucoma**, which can also present with narrow angles and pigment/pseudoexfoliation deposits on the iris. The white lesion could be a pseudoexfoliation flake or pigment deposit. Given the appearance, a narrow angle is a key feature.*

*3. **Evaluate the treatment options:*

*** * **A) Surgery as the first line of treatment:** Surgery (like laser iridotomy or trabeculectomy) is often the definitive treatment for angle-closure glaucoma, especially if the angle is anatomically narrow and causing symptoms or high pressure. However, it's not always the *first* line, especially if the pressure is controlled medically or if the angle closure is intermittent. * **B) Topical corticosteroid drops and dilating drops, with systemic medication when necessary:** Topical corticosteroids are used for inflammation (uveitis), which can sometimes be associated with glaucoma or mimic its symptoms. Dilating drops (mydriatics) are generally *contraindicated* in angle-closure glaucoma because they can further narrow the angle and precipitate an acute attack. Systemic medication might be used for severe inflammation or pressure control, but this combination (especially the dilating drops) is not the standard first-line approach for suspected angle-closure. * **C) No treatment is required:** Glaucoma, especially angle-closure glaucoma, is a serious condition that can lead to irreversible vision loss if untreated. Therefore, no treatment is incorrect. * **D) Laser therapy:** Laser therapy, specifically laser peripheral iridotomy (LPI), is a common and often first-line treatment for *primary angle-closure glaucoma* or *primary angle-closure suspect* to create an alternative pathway for aqueous humor flow and prevent acute attacks. It can also be used in other types of narrow-angle glaucoma.*

*4. **Refine the diagnosis and treatment:** The image strongly suggests a narrow angle, potentially leading to angle-closure glaucoma. The white lesion could be pigment or pseudoexfoliation material. * If it's *primary angle-closure glaucoma* or *primary angle-closure suspect* with a narrow angle, the standard initial treatment is often **laser peripheral iridotomy (LPI)** to prevent acute angle closure. Medical therapy (drops to lower pressure) might be used initially or concurrently, but LPI is frequently the primary intervention to address the underlying anatomical issue. * If it's *secondary*

*angle closure\* due to inflammation (uveitis), then corticosteroids (topical or systemic) would be the primary treatment, along with cycloplegics (which dilate the pupil and can help open the angle in some inflammatory conditions, unlike mydriatics used for examination). \* If it's \*pigmentary glaucoma\* or \*pseudoexfoliation glaucoma\*, treatment focuses on lowering intraocular pressure, often starting with topical medications, but laser trabeculoplasty or surgery might be needed.*

*5. \*\*Re-evaluate the options based on the most likely scenario:\*\* The image most strongly points towards a narrow angle, making angle-closure glaucoma a primary concern. \* Option A (Surgery) is often needed but might not be the absolute \*first\* step if medical therapy or laser therapy can control the pressure and prevent attacks. \* Option B (Corticosteroids, dilating drops) is incorrect because dilating drops are contraindicated in angle closure. \* Option C (No treatment) is incorrect. \* Option D (Laser therapy) is a very common and often first-line treatment (LPI) for preventing acute angle closure in anatomically narrow angles.*

*6. \*\*Compare A and D:\*\* Laser iridotomy (D) is frequently the \*initial\* intervention for primary angle closure or angle-closure suspect to prevent acute attacks. Surgery (A) might be considered later if laser therapy fails or if other conditions are present. Given the options, laser therapy (specifically LPI) is a very strong candidate for the first-line treatment in this scenario.*

*7. \*\*Final Decision:\*\* The image shows features suggestive of angle-closure glaucoma (narrow angle, possible pigment/pseudoexfoliation). Laser peripheral iridotomy is the standard first-line treatment to prevent acute angle closure in such cases. Therefore, Option D is the most appropriate choice among the given options, representing the common initial intervention. \* \*\*Step 1: Analyze the image.\*\* The image shows an eye with a visible iris and pupil. The iris appears somewhat pulled forward, and the angle between the iris and the cornea (anterior chamber angle) seems narrow. There is a white, slightly raised lesion on the iris surface, which could represent pigment dispersion, pseudoexfoliation material, or another deposit. The pupil is relatively round but not fully dilated.\* \*\*Step 2: Identify the likely condition.\*\* The narrow angle appearance is a key feature. This, combined with the possible iris deposits, strongly suggests a condition related to narrow angles, such as primary angle-closure glaucoma (PACG), primary angle-closure suspect (PACS), pigmentary glaucoma, or pseudoexfoliation glaucoma. These conditions involve a physically narrow angle between the iris and cornea, which can impede aqueous humor outflow and lead to increased intraocular pressure (IOP) and potential vision loss. \* \*\*Step 3: Evaluate the treatment options in the context of the likely condition.\*\* \* \*\*A) Surgery as the first line of treatment:\*\* While surgery (like trabeculectomy or tube shunt) is used for glaucoma, it's typically reserved for cases where medical or laser therapy fails, or for specific types of glaucoma. For anatomically narrow angles causing potential angle closure, laser iridotomy is often preferred initially. \* \*\*B) Topical corticosteroid drops and dilating drops, with systemic medication when necessary:\*\* Corticosteroids are used for inflammation (uveitis), which can sometimes cause secondary angle closure, but they are not the primary treatment for primary angle closure. Importantly, \*dilating drops (mydriatics)\* are generally contraindicated in angle-closure glaucoma because they can further narrow the angle and precipitate an acute attack by blocking aqueous outflow through the trabecular meshwork. \* \*\*C) No treatment is required:\*\* Angle-closure glaucoma is a potentially blinding condition that requires treatment to lower IOP and prevent damage to the optic nerve. Therefore, no treatment is incorrect. \* \*\*D) Laser therapy:\*\* Laser peripheral iridotomy (LPI) is a common and often first-line treatment for primary angle-closure glaucoma and angle-closure suspect. It creates a small hole in the peripheral iris, allowing aqueous humor to bypass the narrow angle and flow directly from the posterior to the anterior chamber, thus relieving pupillary block and preventing or treating acute angle closure. Laser trabeculoplasty might be used for open-angle glaucoma or pigmentary/pseudoexfoliation glaucoma, but LPI is the specific laser procedure for narrow angles causing potential closure. \*\*Step 4: Select the most appropriate option.\*\* Given the likely diagnosis of a narrow angle condition potentially leading to angle closure (based on the image), laser peripheral iridotomy (LPI) is the standard initial treatment to prevent acute attacks and manage the condition. Therefore, laser therapy is the most fitting first-line approach among the choices provided.*

*\*\*Final Answer:\*\* The final answer is D."*

Analysis: In this case, the model's CoT reasoning led to an incorrect treatment choice because it misinterpreted the key clinical features emphasized in the annotated CoT. The annotated steps indicate anterior uveitis, characterized by anterior chamber inflammation and posterior synechiae, and recommend topical corticosteroids and dilating drops (Option B). However, the model's reasoning focused

on secondary features such as a narrow anterior chamber angle and iris deposits, overemphasizing the possibility of angle-closure glaucoma, and consequently selected laser therapy (Option D) as the first-line treatment. This illustrates that CoT can sometimes introduce assumptions and reasoning paths that override the correct clinical interpretation, leading to errors even when the direct answer might otherwise be easier to infer.

# E  LIMITATION DISCUSSION

## E.1  ANNOTATION DISCREPANCIES BETWEEN EXPERTS, AI, AND PUBLIC DATASETS

The question-answer pairs and CoT annotations were generated through collaboration between medical experts and AI, while also referencing labels from existing public datasets. In some cases, discrepancies arose between expert judgment and dataset labels. We generally prioritized the public dataset labels as the highest authority. However, we frequently encountered inconsistencies or potential errors in these labels. In such cases, we made efforts to verify through repeated reviews and multiple AI model assessments, but we cannot guarantee that every annotation step is fully accurate.

## E.2  DISEASE-SPECIFIC LABELS MAY IMPLY UNJUSTIFIED DIAGNOSTIC PRECISION

Some annotations involve specific diseases (e.g., COVID-19, certain cancers), directly inherited from the original dataset labels. These labels may have been informed by additional contextual information unavailable in the image alone. In reality, making a definitive diagnosis from a single image is often not feasible, even for trained physicians. By retaining these disease-specific labels, the task may set an unrealistically high bar for MLLMs, possibly exceeding what is expected of human experts. To address this, we aimed to phrase our labels cautiously using formulations like "the most likely diagnosis is...".

## E.3  SUBJECTIVITY IN EXPRESSION MAY AFFECT MATCHING

Although we adopted relatively permissive matching criteria to account for variation in wording, certain annotation statements inevitably involve subjective interpretation, particularly when describing subtle visual findings or formulating likely diagnoses. These subjective elements can introduce variability in phrasing that, despite semantic similarity, may not be captured perfectly by automated matching methods. Furthermore, medical descriptions often allow for multiple valid expressions of the same observation, and differences in terminology, level of detail, or emphasis may lead to mismatches during evaluation. This issue is particularly relevant for open-ended reasoning tasks, where the boundary between correct and incorrect answers can be nuanced.

## E.4  EVALUATION FULLY CONDUCTED WITH MLLMs

All evaluation of model outputs is conducted using GPT-4o, LLaMA-3.3-70B-Instruct-Turbo, and Gemini 2.5 Pro. While these models have demonstrated strong performance in general reasoning and medical question answering, they remain AI systems with inherent limitations. In complex or ambiguous cases, the model may misinterpret medical terminology, overlook subtle differences between options, or apply inconsistent grading criteria. Additionally, its judgments may be influenced by prompt wording or prior context, leading to potential evaluation bias. The absence of human cross-validation may lead to incorrect scores, particularly in domains that require precise domain knowledge.

Regarding circularity concerns, although using a greater variety of models may lead to further improvement, the current annotation workflow is already effective in ensuring high-quality annotations while minimizing model bias. Specifically, by integrating two models, GPT-4o and Gemini 2.5 Pro through multiple processing steps and incorporating manual expert correction, the risk of dominance by a single model has been significantly reduced. Moreover, the final evaluation is based on comparing outputs with the annotated ground truth, rather than relying on the model to generate judgments, further reducing the risk of circularity independently.

### E.5    No Inter-annotator Agreement Scores are Reported

Because this workflow is not fully parallel, we acknowledge that inter-annotator agreement scores are not reported, which is a limitation of this study. However, the multi-stage review process, combining initial student review, multi-model automated checks, targeted expert verification, consensus discussions, and final read-through, ensures high-quality annotations while minimizing bias from any single reviewer or model. This careful workflow allows us to produce reliable reference reasoning chains suitable for evaluating MLLMs in medical image understanding.

### E.6    No multiple experimental runs, and no confidence intervals were reported

Due to cost and time constraints, this study only conducts double evaluations for answer accuracies and CoT steps consistencies, and a single evaluation for the correctness of CoT steps and did not report confidence intervals or significance tests. We acknowledge that repeating experiments and reporting confidence intervals would provide more rigorous and reliable results. In future versions, we plan to include multiple runs and statistical significance analyses to further strengthen the robustness of our findings.

### E.7    Limited Exploration of Prompts and Ablation Studies

In this study, we do not conduct a comprehensive exploration of alternative prompting strategies or perform extensive ablation experiments to evaluate the impact of prompt design choices systematically. Variations such as adjusting the level of detail, explicitly guiding reasoning steps, or introducing domain-specific constraints could potentially influence model performance. Similarly, ablation studies, such as removing specific reasoning cues, altering input formatting, or testing under different context lengths, might have provided more profound insights into model behavior. The absence of these experiments limits our ability to fully characterize how sensitive the results are to prompt engineering and task setup.

## F    Social Impact Discussion

The proposed M3CoTBench benchmark carries several important implications for the development and evaluation of medical AI systems:

### F.1    Advancing interpretable medical AI

By explicitly evaluating the reasoning chains of MLLMs, M3CoTBench encourages transparency in how models arrive at their predictions. Understanding intermediate reasoning steps allows researchers and clinicians to better align AI behavior with clinical decision-making processes, fostering trust and supporting responsible deployment in medical research and practice. In high-stakes medical applications, interpretability is critical: clinicians can verify whether model reasoning is consistent with established diagnostic criteria, and researchers can identify failure modes that may not be apparent from final predictions.

### F.2    Improving model evaluation in medical AI

Most existing benchmarks focus solely on final predictions, overlooking the reasoning process that leads to those outcomes. M3CoTBench fills this gap by providing a structured framework to assess the correctness, consistency, and efficiency of CoT reasoning across diverse medical imaging tasks. This enables a more nuanced analysis of model performance, highlighting specific strengths and weaknesses in reasoning patterns that are essential for complex diagnostic scenarios. By systematically evaluating intermediate steps, M3CoTBench supports the development of models that are not only accurate but also capable of robust and verifiable decision-making.

### F.3    Promoting rigorous development of trustworthy AI systems

By emphasizing the evaluation of reasoning quality rather than only accuracy, the benchmark guides the design of models that are not only correct but also interpretable and reliable. This focus on

transparent reasoning can help mitigate risks associated with opaque AI decisions in clinical settings, enabling more accountable AI deployment. Moreover, by providing standardized metrics for reasoning quality, M3CoTBench encourages best practices in medical AI development, fostering the creation of models that adhere to both technical and ethical standards.

