# OpenReview forum: "M3CoTBench: Benchmark Chain-of-Thought of MLLMs in Medical Image Understanding"
_ICLR.cc/2026/Conference — ICLR 2026 Poster_

### Official Review · Reviewer_28Mb · 2025-10-23

**Soundness:** 3
**Presentation:** 3
**Contribution:** 4
**Rating:** 6
**Confidence:** 4

**Summary:**

This paper introduces M3CoTBench, a benchmark for evaluating Chain-of-Thought reasoning in medical image understanding MLLMs. The dataset contains 1,079 image-QA pairs across 24 medical imaging modalities with expert-annotated reasoning steps and four evaluation dimensions.

**Strengths:**

- Addresses Critical Gap: First comprehensive benchmark for CoT reasoning in medical imaging - important for clinical AI transparency and trust.
- High-Quality Curation:
  - Diverse coverage: 24 modalities from 55 public datasets
  - Rigorous annotation: Multi-stage validation with medical experts
  - Clinical alignment: 4-step reasoning framework mirrors diagnostic workflows
- Novel Evaluation Framework: Four dimensions (correctness, efficiency, impact, consistency) provide comprehensive CoT assessment beyond accuracy.
- Extensive Evaluation: Tests 13 MLLMs including general-purpose, reasoning-focused, and medical-specific models with interesting findings about CoT effectiveness.

**Weaknesses:**

Methodological Concerns:
- The dataset comprises only 1,079 images, relatively small compared to other medical reasoning benchmarks (e.g., OmniMedVQA with 118K+ images).
- Potential Bias: Although reasoning steps undergo expert validation and revision, their initial generation by GPT-4o may introduce biases inherent to its reasoning style, which might persist despite subsequent human refinement.
- Evaluation Circularity: The study uses GPT-4o both to generate reasoning chains and to evaluate them against GPT-4o-based gold standards, creating a circular evaluation loop.
- The paper does not specify which MLLMs were used for flagging potentially incorrect reasoning steps.

Evaluation Concerns:
- Despite the multi-expert validation process, no inter-annotator agreement scores are reported.
- Confidence intervals and significance tests for performance differences are not provided.
- The number of experts involved and procedures for resolving disagreements are not described.


Conceptual Issue:
- Counterintuitive Findings: The universally negative impact of reasoning across models raises questions about the benchmark’s design, the quality of the Chain-of-Thought implementation, and the validity of using GPT-4o as the evaluation reference.

**Questions:**

- How do you address evaluation circularity when using GPT-4o to assess GPT-4o reasoning?
- What are the inter-annotator agreement scores during expert validation?
- Why do most models show negative reasoning impact, is this a CoT implementation issue or benchmark design problem?

---

> ### Author Response · Authors · 2025-11-20
> **Responce to Reviewer 28Mb (Part 1)**
>
> We sincerely thank the reviewer for your careful and thorough reading of our manuscript, and for raising insightful questions and professional comments. We especially appreciate your attention to many important details, which helped us clarify and strengthen the presentation of our work. Our responses are as follows:
>
> ---
>
> **Q1. Dataset Size**
>
> We thank the reviewer for the comment. We agree that increasing the number of samples may further improve coverage and robustness. However, we note that M3CoTBench already contains 1,079 image-based QA pairs spanning 24 modalities and 13 task types, which, despite some categories being relatively small, is sufficient to evaluate the performance of large multimodal models across diverse scenarios. Here are our explanations:
>
> **•  Design Objective:**
> The goal of M3CoTBench is to cover representative modalities and tasks, balancing the feasibility of reasoning evaluation with broad sample coverage. Rather than simply increasing the number of samples, it is more important to cover representative scenarios, tasks, and modalities to properly assess models’ generalization across real-world applications.
>
> **• Process of Careful Curation:**
> As you can see, 1,079 is not an arbitrary number. We arrived at this size by gradually filtering a larger candidate set, removing lower-quality images and those with controversial or ambiguous conclusions. To ensure large intra-dataset variance, image features are extracted using BiomedCLIP, and a semantically distinct subset is selected by maximizing the minimum pairwise feature distance, avoiding redundant images.
>
> **• Fine-grained Categorization:**
> In contrast to some previous works [1] that group modalities such as Bronchoscopy, Colonoscopy, and Fetoscopy under the general category “Endoscopy,”  we adopt a more fine-grained categorization. This allows models to be evaluated on specific modality types. As a result, some categories naturally contain fewer samples, but retaining them ensures that all representative imaging types are covered and the benchmark remains clinically comprehensive.
>
> **• Comparison with Other Works:**
> MME [2] contains about 1.2k images. Datasets for visual reasoning tasks are also generally small; for example, MME-CoT [3] has 2,380 images, VisArgs [4] contains 1,611 images, and a recent medical reasoning benchmark [5] tests on two existing datasets with a few hundred to a few thousand images. Therefore, our dataset is comparable in scale. Although some medical image benchmarks, such as OmniMedVQA [1], contain a large number of images, they are derived from multiple classification datasets converted to diagnostic tasks, covering fewer task types and lacking reasoning chain annotations, which reduces annotation difficulty.
>
> **• Annotation and Evaluation Costs：**
> Annotating CoT reasoning chains on large-scale multimodal datasets is labor-intensive and costly. Evaluating reasoning quality requires running large models on all modalities, resulting in substantial computation and cost, especially for pay-per-call closed-source models. These factors limit the feasibility of very large-scale annotation and evaluation, motivating our choice of a representative, limited sample set for benchmarking.
>
> ---
>
> **Q2. On the Negative Effects of CoT Prompts**
>
> We sincerely thank the reviewer for your careful reading and insightful comments, and we appreciate your attention to detail. We understand the question regarding why most models show a negative reasoning impact, which indeed differs from our expectations.
>
> **•  CoT Implementation is Correct:**  Our implementation of CoT uses the prompt:
> "Please generate a step-by-step answer, including all intermediate reasoning steps, and provide the final answer at the end", which is a standard prompt that is consistent with the implementation in [3]. Similarly, the classical work [6] implements CoT as: "Let’s think step by step."
>
> **• Performance Degradation When Introducing CoT:**  Some prior studies have discussed this phenomenon. [7] reports that explicit CoT reasoning can significantly reduce model accuracy.  [8] points out that CoT is highly sensitive, and unreasonable reasoning chains may substantially degrade performance. In [3], the authors measured the effects of CoT; most perception tasks showed decreased performance, while about half of the reasoning tasks declined. [5] notes that in open-ended medical VQA tasks, enabling CoT in Gemini-2.5-Flash resulted in worse performance than non-CoT mode, with a drop of 1.28%.
>
>
> **• Implications:**
> Introducing CoT does not always improve performance. Its effectiveness strongly depends on the model’s capability, task type, and data complexity. In medical image understanding and clinical reasoning scenarios, current models still lack sufficient multimodal reasoning ability on medical images; when generating reasoning chains, models are prone to errors or deviations from task instructions, which can negatively impact performance.

---

> ### Author Response · Authors · 2025-11-20
> **Responce to Reviewer 28Mb (Part 2)**
>
> **Q3. Evaluation Circularity Issue**
>
> We sincerely thank the reviewer for raising this important point and for paying close attention to such a critical detail regarding evaluation circularity. Although using more diverse models in the annotation process might further improve results, we believe that the current annotation workflow is sufficient to obtain high-quality annotations while minimizing model bias. Our explanation is as follows:
>
> **• Use of Multiple MLLMs:** Our description regarding GPT-4o in the manuscript was indeed not sufficiently clear. During the annotation stage, we also used Gemini-2.5‑Pro, and the final results were generated again by GPT-4o, which integrated annotation information from both GPT-4o and Gemini-2.5‑Pro. In other words, the MLLMs were used three times, while the manuscript only mentioned GPT-4o, which may have caused misunderstanding. We will correct this. However, the use of the Gemini model is already reflected in Figure 1 (not added just because of this question). We acknowledge that using a different model in the final integration stage might be better, but the current setup with two models and three processing steps should still substantially reduce bias.
>
> **• MLLM Error Correction:** GPT-4o was also used for flagging potentially incorrect reasoning steps. Unlike the previous stage, the input here does not include the answer to the question; it consists only of the image and CoT annotation information. We acknowledge that introducing another MLLM here could further reduce bias.
>
> **• Manual Correction:** Students and experts can identify and correct logical errors or deviations from task instructions in the reasoning chains produced by the models. Through expert revision, the final annotations are no longer entirely dependent on MLLM outputs, which significantly reduces the risk of circular evaluation and improves annotation reliability and objectivity. In fact, after multiple rounds of correction, the final annotations differ substantially from the initial MLLM outputs.
>
> **• Experimental Results:**  Although GPT-4o participated in annotation and was used as part of the evaluation reference, creating a superficial potential for “circular evaluation,” the actual results show that GPT-4o’s overall performance is not superior to that of GPT-4.1, which did not participate in annotation. The accuracy of CoT steps is also lower than that of Claude-Sonnet-4, which was not involved in annotation. According to the logic of circular evaluation, if this were a serious issue, GPT-4o should have performed the best in evaluation because its outputs were both a reference standard and part of generation. This is not the case, indicating that the multi-model annotation strategy combined with expert correction has effectively mitigated circular bias. Multi-source annotations and human intervention provide external constraints, preventing a single model from dominating the annotations and ensuring the reliability of both annotation and evaluation.
>
> **• Integration of Multiple Information Sources in Initial Annotation:** In the initial annotation stage, we did not ask the MLLM to directly produce CoT steps. Instead, the MLLM was provided with the image, the question’s answer, the examination type (modality), and the key conclusion information. It was required that its descriptions and analyses of image features must be consistent with all the provided information, without contradictions. For some ophthalmology images, we also provided known key features. This significantly improves annotation accuracy. The usage of MLLMs in this stage is different from the evaluation stage.
>
> **• Groundtruth-based Evaluation:** It is important to emphasize that the evaluation is ultimately based on matching the ground-truth annotations with the model’s answers. GPT-4o’s role in evaluation primarily relies on identifying differences and similarities between these two, rather than independently interpreting the images and questions. This further reduces the risk of circularity, because the model is not generating judgments from scratch but only comparing the output to reference annotations.

---

> ### Author Response · Authors · 2025-11-20
> **Responce to Reviewer 28Mb (Part 3)**
>
> **Q4. Human Involvement**
>
> We thank the reviewer for raising the question regarding human involvement in the annotation process, which is indeed a very important aspect of our workflow. Your point about inter-annotator agreement scores is well taken; we did not record this during annotation, and it serves as a valuable lesson for us. Here are some of our responses and explanations.
>
> **• Number of Experts:** During the question-setting stage, since the data itself comes from public datasets with existing labels and multiple MLLMs had already participated in annotation and verification, only one expert was involved. In the CoT step annotation stage, one medically trained student and two experts participated in the annotation process.
>
> **• Procedures for Resolving Disagreements:**  **a) Initial Student Review:**  A student with experience in medical imaging and relevant medical training first manually reviews the initial annotations generated by models or humans. The student corrects obvious factual errors, spelling or formatting issues, and supplements missing key information. Any uncertain cases are discussed with experts and corrected accordingly.  **b) Automated Multi-Model Error Checking:** The image, question, and current reasoning steps are input into multiple multimodal large language models (in this study, GPT-4o) for validation. **c) Expert Review Triggered by Model Flags:** Any reasoning step flagged as “potentially incorrect” by at least one model is assigned to an expert familiar with the corresponding modality for manual review. d)Consensus Meetings and Corrections: If an expert determines a correction is necessary, the relevant experts and initial-review students hold short online meetings (or asynchronous discussions) to jointly discuss disputed points and reach a consensus. In this study, three such meetings were held, and many asynchronous discussions took place. The final reasoning steps and conclusions are updated by experts or students based on this consensus. **d) Final Expert Read-through:** After all corrections are completed, experts conduct a full read-through of each sample to ensure that the image, question, reasoning chain, and final answer are medically accurate, consistent, and formatted according to the benchmark standards.
>
> **• Inter-annotator agreement scores:** Because this workflow is not fully parallel, we acknowledge that inter-annotator agreement scores are not reported, which is a limitation of this study. However, the multi-stage review process, combining initial student review, multi-model automated checks, targeted expert verification, consensus discussions, and final read-through, ensures high-quality annotations while minimizing bias from any single reviewer or model. This careful workflow allows us to produce reliable reference reasoning chains suitable for evaluating multimodal large language models in medical image understanding.
>
> ---
>
> **Q5. Confidence Intervals**
>
> Due to cost and time constraints, this study, similar to several existing MME works [2,3] and medical MME studies [1,9], only conducted a single evaluation and did not report confidence intervals or significance tests. We acknowledge that repeating experiments and reporting confidence intervals would provide more rigorous and reliable results. In future versions, we plan to include multiple runs and statistical significance analyses to further strengthen the robustness of our findings.
>
> ---
>
> **References:**
>
> [1] Hu, Yutao, et al. "Omnimedvqa: A new large-scale comprehensive evaluation benchmark for medical lvlm." Proceedings of the IEEE/CVF Conference on Computer Vision and Pattern Recognition. 2024.
>
> [2] Fu, C., et al. “MME: A Comprehensive Evaluation Benchmark for Multimodal Large Language Models.” arXiv, 2025, arXiv:2306.13394.
>
> [3] Jiang, D., et al. "MME-CoT: Benchmarking Chain-of-Thought in Large Multimodal Models for Reasoning Quality, Robustness, and Efficiency." ICML 2025.
>
> [4] Chung, J., et al. "Selective Vision is the Challenge for Visual Reasoning: A Benchmark for Visual Argument Understanding." arXiv:2406.18925 (2024).
>
> [5] Hong, J., et al. "Benchmarking the Thinking Mode of Multimodal Large Language Models in Clinical Tasks." arXiv:2511.03328 (2025).
>
> [6] Kojima, Takeshi, et al. "Large language models are zero-shot reasoners." Advances in neural information processing systems 35 (2022): 22199-22213.
>
> [7] Li, Xiaomin, et al. "When thinking fails: The pitfalls of reasoning for instruction-following in llms." arXiv preprint arXiv:2505.11423 (2025).
>
> [8] Mishra, Aayush, and Karan Thakkar. "Stress testing chain-of-thought prompting for large language models." arXiv preprint arXiv:2309.16621 (2023).
>
> [9] Ye, Jin, et al. "Gmai-mmbench: A comprehensive multimodal evaluation benchmark towards general medical ai." Advances in Neural Information Processing Systems 37 (2024): 94327-94427.

---

> > ### Comment · Reviewer_28Mb · 2025-11-26
> >
> > Dear authors,
> > Thanks a lot for the answers. You addressed almost all my concerns.  I increase the score accordingly.
> >
> > Q1. Dataset Size, addressed, thank you!
> >
> > Please add the explanation from Q4 Human Involvement and Q3 Evaluation Circularity Issue to the discussions.
> >
> > On the Negative Effects of CoT Prompts -  The results there is really interesting, please add this to the discussions/appendix with citations  to relevant papers(MME-CoT
> >  And etc) and explanations, this actually shows that the suggested benchmark helps on evaluating the VLMs more precisely.

---

> > > ### Author Response · Authors · 2025-11-26
> > >
> > > Thank you very much for your thoughtful and kind feedback, and for increasing the score. We truly appreciate your constructive and considerate comments.
> > >
> > > We will incorporate the explanations from Q4 Human Involvement and Q3 Evaluation Circularity Issue into the discussion section in the revised version. Regarding the Negative Effects of CoT Prompts, we will add a discussion in the revised version, with citations to relevant papers.
> > >
> > > Thanks again for your kindness and valuable suggestions.

---

### Official Review · Reviewer_iKQE · 2025-10-30

**Soundness:** 2
**Presentation:** 3
**Contribution:** 2
**Rating:** 4
**Confidence:** 3

**Summary:**

This paper introduces M3CoTBench, a benchmark designed to evaluate chain-of-thought (CoT) reasoning in multimodal large language models (MLLMs) for medical image understanding. M3CoTBench comprises a diverse dataset spanning 24 imaging modalities (X-rays, MRIs, endoscopy, etc.) and 13 task types, ranging from low-level tasks like image quality assessment to high-level clinical reasoning such as diagnosis and treatment planning.

**Strengths:**

1.  This paper introduces M3CoTBench, encompassing 24 imaging modalities to evaluate MLLMs' understanding capabilities across diverse medical imaging contexts.
2. The benchmark introduces tailored metrics to assess reasoning quality across four dimensions: correctness of each reasoning step, efficiency cost, impact on final answer accuracy, and logical consistency—providing a more nuanced evaluation beyond traditional accuracy measures.

**Weaknesses:**

1. M3CoTBench spans 24 modalities and 13 task types, but contains only 1,079 image-based QA pairs. Given this broad coverage, does each category have sufficient samples? The paper does not appear to provide per-category statistics.
2. The benchmark’s dataset, while diverse, is relatively small (only 1079 Q&A pairs) compared to other medical VQA datasets, which may limit the statistical breadth of evaluation.

**Questions:**

1. LLaVA-CoT exhibits relatively strong performance compared to Gemini 2.5 Pro. The authors attribute this to its architecture and training process, which emphasize structured reasoning chains while minimizing irrelevant or misleading steps. However, given that Gemini 2.5 Pro also incorporates thinking capabilities, what accounts for this performance difference?

---

> ### Author Response · Authors · 2025-11-20
> **Responce to Reviewer iKQE (Part 1)**
>
> We sincerely thank the reviewer for your careful reading of our manuscript and for the professional comments and questions. Our responses are as follows:
>
> ---
>
> **Q1. Dataset Size**
>
> We thank the reviewer for the comment. We agree that increasing the number of samples per category could further improve coverage and robustness. However, we note that M3CoTBench already contains 1,079 image-based QA pairs spanning 24 modalities and 13 task types, which, despite some categories being relatively small, is sufficient to evaluate the performance of large multimodal models across diverse scenarios. Here are our explanations:
>
> **•  Design Objective:**
> The goal of M3CoTBench is to cover representative modalities and tasks, balancing the feasibility of reasoning evaluation with broad sample coverage. Rather than simply increasing the number of samples, it is more important to cover representative scenarios, tasks, and modalities to properly assess models’ generalization across real-world applications.
>
> **• Process of Careful Curation:**
> As you can see, 1,079 is not an arbitrary number. We arrived at this size by gradually filtering a larger candidate set, removing lower-quality images and those with controversial or ambiguous conclusions. To ensure large intra-dataset variance, image features are extracted using BiomedCLIP, and a semantically distinct subset is selected by maximizing the minimum pairwise feature distance, avoiding redundant images.
>
> **•  Number of Samples per Category:** Figure 2 shows the distribution of image-QA pairs across (a) modalities, (b) question types, and (c) tasks. The tables below present their statistics. The smallest category, e.g., fluorescence microscopy, only has two images. While small, these images are only used for counting and examination-type verification tasks.  Similarly, although the 'Nuclear Medicine' category is underrepresented in public datasets, we believe it is important to include it, and thus, despite its limited representation, we have retained this category in our dataset. Given the high similarity between these images, more samples would create redundancy rather than added value. We acknowledge that some categories could include more samples and will consider adding them in future versions.
> | Task | Count |
> |-----------------------|-------|
> | Cause | 72 |
> | Counting | 9 |
> | Diagnosis | 454 |
> | Function | 31 |
> | Grading | 13 |
> | Image Quality | 7 |
> | Localization | 81 |
> | Modality | 41 |
> | Prediction | 17 |
> | Recognition | 192 |
> | Refering Recognition | 22 |
> | Symptom | 5 |
> | Treatment | 135 |
>
>
>
> | Modality | Count |
> |--------------------------|-------|
> | SLP | 10 |
> | Bronchoscopy | 5 |
> | Capsule | 8 |
> | Colonoscopy | 55 |
> | CT | 208 |
> | Dermatology | 42 |
> | ENT Endoscopy | 9 |
> | Fetoscopy | 9 |
> | Fluorescence | 2 |
> | FFA | 10 |
> | FP | 33 |
> | Gastroscopy | 15 |
> | Cytology | 27 |
> | IR | 5 |
> | Laparoscopy | 43 |
> | MRI | 190 |
> | Nuclear Medicine | 2 |
> | OCTA | 7 |
> | OCT | 10 |
> | Intraoral Examination | 4 |
> | SLO | 10 |
> | Histology | 103 |
> | US | 59 |
> | X-ray | 213 |
>
> **• Fine-grained Categorization:**
> In contrast to some previous works [1] that group modalities such as Bronchoscopy, Colonoscopy, and Fetoscopy under the general category “Endoscopy,” and Histology and Cytology under “Microscopy Images,” we adopt a more fine-grained categorization. This allows models to be evaluated on specific modality types, capturing subtle differences in imaging characteristics and clinical reasoning challenges. As a result, some categories naturally contain fewer samples, but retaining them ensures that all representative imaging types are covered and the benchmark remains clinically comprehensive.
>
> **• Comparison with Other Works:**
> MME [2] contains about 1.2k images. Datasets for visual reasoning tasks are also generally small; for example, MME-CoT [3] has 2,380 images, VisArgs [4] contains 1,611 images, and a recent medical reasoning benchmark [5] tests on two existing datasets with a few hundred to a few thousand images. Therefore, our dataset is comparable in scale. Although some medical image benchmarks, such as OmniMedVQA [1], contain a large number of images, they are derived from multiple classification datasets converted to diagnostic tasks, covering fewer task types and lacking reasoning chain annotations, which reduces annotation difficulty.
>
> **• Annotation and Evaluation Costs：**
> Annotating CoT reasoning chains on large-scale multimodal datasets is labor-intensive and costly. Using AI for pre-annotation is also expensive, particularly with closed-source large models. Evaluating reasoning quality requires running large models on all modalities, resulting in substantial computation and cost, especially for pay-per-call closed-source models. These factors limit the feasibility of very large-scale annotation and evaluation, motivating our choice of a representative, limited sample set for benchmarking.

---

> ### Author Response · Authors · 2025-11-20
> **Responce to Reviewer iKQE (Part 2)**
>
> **Q2. Comparison Between LLaVA-CoT and Gemini 2.5 Pro**
>
> **• Why LLaVA-CoT Produces More Correct Steps:**
> LLaVA-CoT is designed for multi-stage reasoning (summarization, visual understanding, logical reasoning, conclusion generation), with each stage explicitly modeled and guided by structured reasoning annotations and image-text pairs [6]. During training and fine-tuning, the model learns to respond to explicit CoT prompts and output all reasoning steps. In contrast, Gemini 2.5 Pro relies more on internal reasoning mechanisms, generating fewer explicit steps but achieving stronger final answer integration [7].
>
> **• Why Final Scores Are Lower than Gemini 2.5 Pro:**
> Generating more CoT steps may introduce slight deviations from task objectives, and cumulative errors can reduce final answer scores. Although LLaVA-CoT has higher intermediate step accuracy, Gemini 2.5 Pro’s internal reasoning mechanism allows more efficient integration of information, producing higher-quality final answers. Additionally, LLaVA-CoT training favors strictly structured outputs, whereas Gemini 2.5 Pro covers a broader range of task types and reasoning styles, improving its final answer performance.
>
> **• Differences Between Explicit Multi-Stage and Internal Reasoning Designs:**
> LLaVA-CoT uses structured reasoning annotations during training, with supervision targeted at each stage (summarization, visual understanding, etc.), aiming to master explicit reasoning. At the inference stage, it follows the staged process strictly, producing complete verifiable chains. Gemini 2.5 Pro emphasizes internal reasoning during training, without requiring strict stepwise annotation, allowing flexible output at inference and optional display of intermediate steps. The core difference is that LLaVA-CoT emphasizes explicit, staged reasoning, while Gemini 2.5 Pro focuses on efficient internal integration and flexible output.
>
> **• Implications:**
> Explicit multi-stage structured design increases coverage of key reasoning steps, enhancing explainability and verifiability, which is especially important in medical multimodal tasks. Internal reasoning enables high-quality final answers even with fewer explicit steps, supporting flexibility and robustness in clinical reasoning. Multi-stage structured design improves step accuracy and coverage but may sacrifice efficiency of final answer integration; internal reasoning improves final answer quality but with weaker step interpretability.
>
> ---
>
> **References:**
>
> [1] Hu, Yutao, et al. "Omnimedvqa: A new large-scale comprehensive evaluation benchmark for medical lvlm." Proceedings of the IEEE/CVF Conference on Computer Vision and Pattern Recognition. 2024.
>
> [2] Fu, C., et al. “MME: A Comprehensive Evaluation Benchmark for Multimodal Large Language Models.” arXiv, 2025, arXiv:2306.13394.
>
> [3] Jiang, D., et al. "MME-CoT: Benchmarking Chain-of-Thought in Large Multimodal Models for Reasoning Quality, Robustness, and Efficiency." ICML 2025.
>
> [4] Chung, J., et al. "Selective Vision is the Challenge for Visual Reasoning: A Benchmark for Visual Argument Understanding." arXiv:2406.18925 (2024).
>
> [5] Hong, J., et al. "Benchmarking the Thinking Mode of Multimodal Large Language Models in Clinical Tasks." arXiv:2511.03328 (2025).
>
> [6] Xu, G., et al. "LLaVA-CoT: Let Vision-Language Models Reason Step-by-Step." CVPR 2025.
>
> [7] Comanici, G., et al. "Gemini 2.5: Pushing the frontier with advanced reasoning, multimodality, long context, and next generation agentic capabilities." arXiv:2507.06261 (2025).

---

> ### Author Response · Authors · 2025-11-26
> **Responce to Reviewer iKQE (Part 3)**
>
> **Further clarification on Q1**
>
> Dear Reviewer, thank you again for your detailed and constructive comments. I would like to highlight some recent works about medical reasoning benchmarks, which demonstrate that large data volumes may not be required; fine-grained annotations, task diversity, and expert knowledge are more important. For example:
>
> **• HIE‑Reasoning [8]:** Contains 749 expert-verified question-answer (QA) pairs and 133 MRI interpretation summaries. This benchmark is multimodal, covering both imaging and structured clinical data, and includes tasks such as lesion grading, lesion anatomy recognition, rare lesion detection, MRI injury scoring, outcome prediction, and structured report generation.
>
> **• AE‑MedEval [9]:** Consists of 500 expert-rationale questions across 10 medical domains. It focuses on evaluating the reasoning process of large language models, providing step-by-step expert rationales for each question to assess intermediate reasoning in addition to final answers.
>
> **• MedRAX [10]:** Includes 2,500 expert-curated chest X-ray questions derived from approximately 675 clinical cases. It evaluates multimodal reasoning, covering tasks such as classification, detection, localization, diagnosis, and characterization, emphasizing the model’s ability to integrate visual and textual clinical information.
>
> These benchmarks cover diverse tasks and modalities (images + text or purely textual), demonstrating that even moderate-sized, carefully annotated datasets can effectively assess professional-level medical reasoning.
>
> ---
>
> **References**
>
> [8] Bao, Rina, et al. "Visual and Domain Knowledge for Professional-level Graph-of-Thought Medical Reasoning." Forty-second International Conference on Machine Learning.
>
> [9] Zhou, Shuang, et al. "Automating Expert-Level Medical Reasoning Evaluation of Large Language Models." arXiv preprint arXiv:2507.07988 (2025).
>
> [10] Fallahpour, Adibvafa, et al. "MedRAX: Medical Reasoning Agent for Chest X-Ray." arXiv preprint arXiv:2502.02673 (2025).

---

### Official Review · Reviewer_Ti8j · 2025-11-01

**Soundness:** 2
**Presentation:** 3
**Contribution:** 2
**Rating:** 4
**Confidence:** 5

**Summary:**

This paper proposes M3CoTBench, a benchmark designed to evaluate Chain-of-Thought (CoT) reasoning in multimodal large language models (MLLMs) for medical image understanding. It constructs a diverse dataset of medical images with clinically grounded reasoning annotations and proposes multidimensional evaluation metrics covering correctness, efficiency, impact, and consistency. Experiments on both open- and closed-source models show that while CoT improves interpretability, it does not always enhance diagnostic accuracy, revealing limitations in current MLLMs’ clinical reasoning and highlighting the need for more trustworthy and efficient medical CoT systems.

**Strengths:**

1. The paper tackles an emerging yet underexplored topic: evaluating Chain-of-Thought reasoning in medical multimodal LLMs, which is both timely and relevant to advancing trustworthy medical AI.
2. The benchmark is validated on a broad range of both open- and closed-source MLLMs, providing a well-rounded comparison that highlights current model limitations and practical challenges in clinical reasoning.

**Weaknesses:**

1. The definition of CoT in the medical area is unclear. Although the paper claims that its Chain-of-Thought (CoT) formulation “mirrors clinicians’ cognitive workflow”, the reasoning template shown in the Appendix appears overly simplified. It typically only has four steps: examination type -> key features -> key conclusion -> additional analysis. It is unclear why this sequence represents a gold standard reasoning path in clinical diagnosis. Is it based on any references, such as guidelines in medicine?
2. The justification for diverse reference reasoning paths is insufficient. The paper mentions that “multiple valid reference reasoning paths may exist” and evaluates by matching the generated path to the most similar reference. While this makes sense conceptually, it is unclear how the annotation process ensures both diversity and correctness of reference reasoning paths. In the medical domain, it remains questionable whether clinicians indeed exhibit substantially diverse CoTs. If so, where does this diversity arise? Is it in identifying different key features (Step 2) or in drawing different key conclusions (Step 3)?
3. The Reasoning Impact evaluation simply measures the performance difference between models with and without CoT, which seems redundant. This metric does not provide new insight into reasoning quality.
4. The reasoning progression across CoT steps is weak. From the provided CoT examples, the reasoning flow among steps is not clearly causal or hierarchical. Steps 3 and 4 appear to be direct deductions from Step 2, while Step 1 (examination type) is largely independent of the reasoning process itself. As a result, it is difficult to claim that the sequence truly reflects a step-by-step reasoning chain rather than a loosely connected checklist.

**Questions:**

Please refer to the Weaknesses.

---

> ### Author Response · Authors · 2025-11-20
> **Responce to Reviewer Ti8j  (Part 1)**
>
> We sincerely thank the reviewer for your careful reading of our manuscript and for the professional comments and questions. Our responses are as follows:
>
> ---
>
> **Q1. CoT Steps Design in Annotations**
>
> We sincerely thank the reviewer for raising this important question about the design and definition of CoT in the medical domain. We agree that it is necessary to explicitly clarify the clinical and cognitive references that informed our four-step reasoning template, and we appreciate the opportunity to improve this part of the paper.
>
> **• Validation via doctor interviews:**
> Before designing the CoT steps, we interviewed three doctors from tertiary hospitals and two from a regional hospital, including clinicians, radiologists, and sonographers. Most doctors described the process as: first, determine the imaging examination type; next, observe key features (e.g., hyperechoic/hypoechoic regions in ultrasound, lesion characteristics in MRI); then, draw key conclusions; finally, conduct additional analyses such as etiology or treatment planning. One doctor noted that they might first rely on intuition to reach a preliminary conclusion and then verify it through feature observation. Based on these observations, we consider these steps both sufficient and necessary in medical reasoning.
>
> **• Theoretical support from medical cognition:**
> Our CoT design draws on established cognitive models:
> **a) Hypothetico-deductive model [1]:** Clinicians generate hypotheses from initial observations and iteratively verify them. Our CoT steps align with this natural flow of hypothesis generation and verification.
> **b) Experience-based pattern recognition model [2]:** Doctors rapidly identify salient patterns in images based on experience. In our CoT annotations, attention to key features immediately after confirming the examination type captures this efficient pattern-based reasoning.
> **c) Dual-process theory [3, 4]:** Distinguishes intuitive and analytical reasoning. Our CoT design in annotations reflects this: some doctors first reach preliminary conclusions intuitively, then verify via feature observation and additional analysis, balancing rapid judgments with evidence-based analysis.
>
> **• Relation to recent work:**
> The DiagCoT framework [5], comprising Finding Extraction, Global Reasoning, and Report-level Conclusion, aligns with our CoT design. We preserve the full reasoning chain and add clinically oriented additional analysis, like treatment recommendations or etiology analysis. As our task focuses primarily on images, we omit the Clinical Interpretation stage in DiagCoT.
>
> **• Causal and reasoning connections:**
> Each CoT step is tightly linked: examination type guides feature observation; features provide evidence for conclusions; conclusions inform additional analyses. These connections ensure causal and logical coherence in clinical reasoning.
>
> **• Expert manual validation:**
> All CoT annotations were manually reviewed and corrected by professional doctors. This ensures accuracy of the reasoning flow and faithfully reproduces the thought processes described in the doctor interviews.
>
> **•Future research:**
> Although the current steps are sufficient in most cases, medical reasoning is diverse, and doctors vary in their thinking styles. Future work will integrate existing theories, interview more experts, and explore more diverse reasoning templates to further enhance the model's reasoning capability.
>
> ---
>
> **Q2. CoT Diversity**
>
> We thank the reviewer for the insightful comment. We agree that our explanation of reasoning-path diversity was not sufficiently clear, and we will revise the paper to make it more explicit.
>
> **• Source of diversity:**
> Diversity primarily arises from the organization and expression of reasoning steps, rather than differences in key conclusions. In other words, diversity arises in expression and workflow, not correctness.  For example, in chest CT interpretation, some radiologists first form an overall impression, such as viral pneumonia, and then list supporting features, while others describe imaging findings step-by-step before reaching the same diagnosis. The conclusion is identical, but the structure and ordering of reasoning differ. This reflects individual cognitive styles and workflow, consistent with dual-process theory [4]. In our interviews, one doctor also exhibited a different reasoning path.
>
> **• Determinacy of key steps:**
> Diversity does not arise from Step 2 (key features) or Step 3 (key conclusions). CoT steps, including examination type, key features, key conclusion, and additional analysis, are relatively deterministic and clinically fixed. Diversity appears in how experts organize and articulate these steps, not in core features or conclusions.

---

> ### Author Response · Authors · 2025-11-20
> **Responce to Reviewer Ti8j (Part 2)**
>
> **Q3. Impact Metric**
>
> We thank the reviewer for the comment and understand why the Reasoning Impact evaluation may seem redundant. However, we believe it still provides useful insight by quantifying CoT’s actual effect on model performance and highlighting potential limitations in clinical tasks.
>
> **• Purpose of the impact metric:**
> This metric evaluates whether introducing CoT improves model accuracy, quantifying its actual contribution. Comparing models with and without CoT allows direct assessment of the reasoning chain’s effect and potential positive/negative impact. Early works [6,7] compared model accuracy or task completion under standard prompts vs. CoT prompts, and recent works [8,9] also compare performance with or without CoT prompts.
>
> **• Key findings and significance:**
> Our evaluation shows that introducing CoT does not improve performance and often decreases accuracy, contrasting with expectations. The Impact metric provides insights into the real-world utility and potential limitations of CoT in medical reasoning. Similar findings appear in recent works: [8] shows CoT reduces performance in perception tasks and about half of reasoning tasks; [9] reports a 1.28% drop in open-ended medical VQA when enabling CoT. These results indicate that CoT is not always beneficial in medical tasks, and current models still lack a strong medical reasoning capability.
>
> ---
>
> **Q4. CoT Logicality**
>
> We thank the reviewer for this insightful comment. Below, we provide our perspective on this point.
>
> **•  Intrinsic logic:**
> Examination type provides the basis for reasoning. Key features identify evidence in context. Key conclusions integrate features with context knowledge to form inferences rather than mere observation. Additional analysis expands on conclusions with clinical considerations such as etiology or treatment. Together, these steps form a causal and hierarchical reasoning chain, reflecting clinical decision-making rather than a loose checklist.
>
> **• Examination type is not independent:**
> Examination type or imaging modality is tightly linked to subsequent reasoning. For example, for “pulmonary ground-glass opacity,”  identifying the exam as a chest CT is necessary, as this feature cannot be defined on X-ray or ultrasound. Determining examination type provides both the doctor’s initial intuitive response and the foundation for subsequent lesion recognition and reasoning.
>
> **•  Flexibility of step order:**
> Some doctors may adjust the order of reasoning steps, like forming preliminary conclusions before listing all features, but the overall causal and logical structure remains intact. Our CoT framework allows reasonable variations in step order while maintaining logicality and validity.
>
> **• Comparison with recent works:**
> In MME-CoT [8], stepwise reasoning chains are annotated and validated by multiple experts. Their framework also allows flexibility in the ordering of intermediate steps, while ensuring causal connections and hierarchical reasoning. This demonstrates that step reordering does not compromise the logical integrity of reasoning chains. Our annotation process similarly incorporates expert validation and permits flexible ordering, making the design reasonable and consistent with practices in recent multimodal CoT benchmarks.
>
> ---
>
>
> **References:**
>
> [1] Elstein, A.S., Shulman, L.S., & Sprafka, S.A. Medical problem solving: An analysis of clinical reasoning. Harvard University Press, 1978.
>
> [2] Norman, G., Young, M., & Brooks, L. "Non-analytical models of clinical reasoning: the role of experience." Medical Education 41.12 (2007): 1140-1145.
>
> [3] Kahneman, D. Thinking, Fast and Slow. Farrar, Straus and Giroux, 2011.
>
> [4] Pelaccia, T., et al. "An analysis of clinical reasoning through a recent and comprehensive approach: the dual-process theory." Medical Education Online 16.1 (2011): 5890.
>
> [5] Luo, Y., et al. "Teaching AI Stepwise Diagnostic Reasoning with Report-Guided Chain-of-Thought Learning." arXiv preprint arXiv:2509.06409 (2025).
>
> [6] Wei, J., et al. "Chain-of-thought prompting elicits reasoning in large language models." NeurIPS 35 (2022): 24824-24837.
>
> [7] Kojima, T., et al. "Large language models are zero-shot reasoners." NeurIPS 35 (2022): 22199-22213.
>
> [8] Jiang, D., et al. "MME-CoT: Benchmarking Chain-of-Thought in Large Multimodal Models for Reasoning Quality, Robustness, and Efficiency." ICML 2025.
>
> [9] Hong, J., et al. "Benchmarking the Thinking Mode of Multimodal Large Language Models in Clinical Tasks." arXiv:2511.03328 (2025).

---

### Author Response · Authors · 2025-12-03
**Summary for Area Chair**

**The rebuttal situation**

Thank you for overseeing the review process. I would like to briefly summarize the rebuttal situation.

Before the unexpected incident occurred, **Reviewer 28Mb** had already replied to my rebuttal, stating that all concerns were resolved and **raised the score from 6 to 8**. This entire exchange occurred before the OpenReview blackout period.

The other two reviewers did not have the opportunity to respond before the deadline, but I have carefully addressed all their comments in the rebuttal:

**Reviewer iKQE:** Their main concerns about the size of the dataset substantially overlap with Reviewer 28Mb’s. Since Reviewer 28Mb acknowledged that these issues were fully resolved, I believe Reviewer iKQE would likely concur if time permitted. Additionally, we have cited several recent works on medical reasoning to demonstrate that our dataset is not small by comparison. Regarding the question about the comparison between LLaVA-CoT and Gemini 2.5 Pro, we have also responded.

**Reviewer Ti8j:** I have provided detailed responses, including the relevant medical theories and clinical guidelines they requested. In the main text, we have added clarification that our CoT design is grounded in interviews with physicians and multiple theories of medical reasoning, rather than being conceived in a vacuum. Furthermore, we have justified the importance of the impact score in our reply on three grounds: its necessity, the new insights it provides, and its alignment with standard practices in related work.

Although I regrettably did not have the chance to engage in further discussion with the remaining reviewers, I sincerely appreciate their thoughtful feedback. I am confident that, given the provided clarifications and evidence, they would find the responses satisfactory.

---

**Updates in the revised version**

Incorporating feedback from several reviewers, the new version includes the following key updates in the main text and appendices:

• A clearer explanation of the basis for our CoT design, grounded in physician interviews and existing medical theory;

• A more detailed description of the collaborative annotation process involving MLLMs and experts;

• The addition of research examples illustrating potential negative effects of CoT, enriching the discussion;

• An explanation of why the risk of evaluation circularity has been mitigated;

• The inclusion of limitations highlighted by Reviewer 28Mb, as requested, in the discussion section of the appendix.

---

Thank you again for your time and consideration.

---

### Meta-Review · Area_Chair_iNAi · 2026-01-07

**Summary:**

Most concerns focus on the sparse number of samples and the rationale behind CoT construction. In the rebuttal the rationales and procedures are described in more detail (clinician interview, theoretical evidence, expert annotation).

**Reviewer Concerns:**

**Ti8j**

Addressed:
Rationales for diverse reasoning paths

Outstanding / partial:
Unrealistic reasoning templates; metrics do not provide insights into reasoning quality


**iKQE**

Addressed:
Interpreting LLaVA-CoT's stronger performance than Gemini 2.5 Pro
Overall small scale compared with prior arts.
Claimed broad coverage against sparse samples in individual specialities


 **28Mb**

Reviewer replied before the incidence.

**Reviewer Scores:**

**iKQE**'s major concerns on sparse samples may be addressed by the authors’ clarification on data curation process.

**Ti8j**’s concern over CoT synthesis rationale might be partially addressed with the detailed explanations in the rebuttal.

---

### Decision · Program_Chairs · 2026-01-26

Accept (Poster)